# Coupled Reconstruction of Cortical Surfaces by Diffeomorphic Mesh Deformation

**Hao Zheng     Hongming Li     Yong Fan**
University of Pennsylvania
hzheng1@upenn.edu, {hongming.li,yong.fan}@pennmedicine.upenn.edu

## Abstract

Accurate reconstruction of cortical surfaces from brain magnetic resonance images (MRIs) remains a challenging task due to the notorious partial volume effect in brain MRIs and the cerebral cortex's thin and highly folded patterns. Although many promising deep learning-based cortical surface reconstruction methods have been developed, they typically fail to model the interdependence between inner (white matter) and outer (pial) cortical surfaces, which can help generate cortical surfaces with spherical topology. To robustly reconstruct the cortical surfaces with topological correctness, we develop a new deep learning framework to jointly reconstruct the inner, outer, and their in-between (midthickness) surfaces and estimate cortical thickness directly from 3D MRIs. Our method first estimates the midthickness surface and then learns three diffeomorphic flows jointly to optimize the midthickness surface and deform it inward and outward to the inner and outer cortical surfaces respectively, regularized by topological correctness. Our method also outputs a cortex thickness value for each surface vertex, estimated from its diffeomorphic deformation trajectory. Our method has been evaluated on two large-scale neuroimaging datasets, including ADNI and OASIS, achieving state-of-the-art cortical surface reconstruction performance in terms of accuracy, surface regularity, and computation efficiency.

## 1   Introduction

The analysis of the cerebral cortex using magnetic resonance imaging (MRI) is crucial in understanding neurodegenerative diseases [9, 43] and psychological disorders [42]. Since the cerebral cortex is a thin (a few millimeters thick) and highly folded sheet between the inner (white matter: WM) and outer (pial) surfaces, voxel-based segmentation methods cannot accurately capture its complicated morphology [17]. Instead, triangular meshes have been widely used for cortical surface reconstruction (CSR) [14] in order to accurately measure cortical thickness, volume, and gyrification. Although well-established methods for CSR (e.g., FreeSurfer [17], BrainSuite [48]) can produce promising results, they often require significant computational resources (e.g., 6h/subject [17]) and may necessitate manual editing in order to attain sub-voxel precision.

Recently, deep learning (DL) methods have achieved significant improvement in CSR in terms of both accuracy (sub-voxel error) and efficiency (orders of magnitude faster) [10, 13, 19, 20, 26, 30, 31, 41, 47, 59]. These DL methods can be broadly classified into two categories according to the representation of the output surfaces. (I) Implicit surface representation, such as signed distance function [13, 19], occupancy filed [13], and level set [41], can be predicted by neural networks, and cortical surface is then generated by a marching cubes algorithm [27]. (II) Explicit surface reconstruction methods take a coarse or fine initial mesh as input and directly predict a target mesh [10, 20, 26, 30, 31, 59]. A detailed comparison of the existing DL-based CSR methods is summarized in Table 1.

37th Conference on Neural Information Processing Systems (NeurIPS 2023).

Table 1: A comparison of existing DL-based CSR methods and ours. For input, "volume": the whole 3D volume; "cube": a sub-volume; FS: fine surface mesh; CS: coarse surface template; IS: implicit surface representation. For output, explicit surface: triangular mesh; implicit surface: signed distance function or occupancy field.

| Methods | Input | | Network | | Output | Joint surf. reconstruction | Direct thickness estimation |
|---|---|---|---|---|---|---|---|
| | Img. | Surf. | CNN | GNN/MLP | Explicit surf. | | |
| DeepCSR [13] | volume | | ✓ | ✓ | | | |
| vox2surf [19] | vol.+cube | | ✓ | ✓ | | | |
| FastCSR [41] | volume | | ✓ | | | | |
| PialNN[31] | cube | FS | ✓ | ✓ | ✓ | | |
| CorticalFlow [26, 47] | volume | CS | ✓ | | ✓ | | |
| vox2cortex [10] | volume | CS | ✓ | ✓ | ✓ | ✓ | |
| TopoFit [20] | volume | CS | ✓ | ✓ | ✓ | | |
| CortexODE[30] | cube | FS | ✓ | ✓ | ✓ | | |
| Ours | volume | FS+IS | ✓ | | ✓ | ✓ | ✓ |

However, the existing *DL-based CSR* methods are subject to limitations. ***First***, the *interdependence* between the inner and outer cortical surfaces is generally ignored, and therefore separate or multi-stage DL models are typically trained to reconstruct both the inner and outer cortical surfaces and intersections between them may occur. Even if both the inner and outer surfaces can be reconstructed simultaneously [10], they are loosely combined with no topological constraints. ***Second***, complex DL architectures are commonly used, with *separate* learning of image and surface/vertex features using both convolutional neural networks (CNNs) and graph neural networks (GNNs)/multi-layer perceptrons (MLPs). Moreover, graph convolutions become less scalable as the number of vertices in the mesh grows to accommodate complex shapes, and often fail to learn diffeomorphic mappings to produce genus-zero regular meshes [10]. ***Third***, using a coarse mesh template incurs difficulty of learning large deformations for highly folded cortical regions and may lead to non-smooth deformation and undesirable artifacts. ***Last but not least***, the cortical thickness estimation is neglected in *all* existing *DL-based CSR* methods (i.e., needs a separate step to compute cortical attributes as in conventional pipelines [17, 56]). It may serve as an anatomical constraint to couple the inner and outer cortical surfaces, help generate cortical surfaces with topological correctness, and facilitate quantitative analysis of the cortical thickness.

In order to robustly reconstruct the cortical surfaces with topological correctness, we develop a DL-based approach to simultaneously reconstruct both the inner and outer cortical surfaces and estimate the cortical thickness by optimizing and deforming an initialized midthickness surface. ***First***, our method explicitly couples the inner and outer surfaces by jointly learning three diffeomorphic flows to optimize the initialization midthickness surface to lie halfway between the inner and outer cortical surfaces and deform it to inner and outer cortical surfaces, respectively. ***Second***, instead of designing a complex mixed architecture of CNNs and GNNs/MLPs, our method employs a single model of 3D CNNs to predict the diffeomorphic flows from a multi-channel input, consisting of a 3D brain MRI, a ribbon segmentation map that encodes structural information of the cerebral cortex, and a signed distance function that implicitly encodes the initialization surface. ***Third***, our method calculates the diffeomorphic deformation trajectories in a continues coordinate space rather than on a 3D voxelwise grid, achieving higher sub-voxel accuracy from fine-grained velocity fields while maintaining reasonable computational efficiency even as the number of mesh vertices increases. ***Fourth***, we devise an efficient and reliable approach to initialize *a fine midthickness surface* from a cortical ribbon segmentation result, followed by topology correction to ensure genus zero. ***Finally***, a vertex-wise thickness estimation can be obtained by tracing the geodesic trajectory of each vertex during the mesh deformation process. *In summary*, our new DL framework differs from the existing DL-based CSR approaches in its coupled reconstruction of multiple surfaces and the simultaneous cortical thickness estimation, which facilitates robust reconstruction of the cortical surfaces with spherical topology. Ablation studies and comparison experiments on two large public datasets (ADNI [22] and OASIS [33]) have demonstrated that our method attains superior performance over the state-of-the-art methods [10, 13, 26, 30, 31, 47].

## 2 Related Works

### 2.1 Learning-based Cortical Surface Reconstruction

Recent years have witnessed a surge of interest in geometric DL-based methods for general computer vision tasks [15, 29, 49, 52, 54, 57, 58]) and biomedical object reconstruction [24, 25, 37, 38, 55].

However, their applications to biomedical tasks are limited to organs with simple shapes such as the liver and heart. The cerebral cortex, on the other hand, is a highly folded, thin structure with a significantly complex shape, necessitating more advanced approaches.

Table 1 summarizes two categories of CSR methods, i.e., implicit and explicit surface reconstruction. The implicit surface reconstruction methods typically learn a function that maps 3D coordinates to a continuous implicit representation of the surface, such as signed distance function [39] and occupancy field [34]. Compared to well-established pipelines [17], these methods have significantly improved inference efficiency (reducing time cost from hours to minutes). Despite their ability to generate surfaces at any desired resolutions [13, 18, 19, 41], these methods often rely on a marching cubes algorithm [27] to obtain a triangular mesh and post-processing topology correction methods [7] for ensuring correct spherical topology of the reconstructed surfaces.

The explicit surface reconstruction methods typically learn a mapping from an initialization mesh to a target surface and the mapping can be modeled as a deformation model [52, 53, 55]. Particularly, Pixel2Mesh [52] utilizes GNNs to learn vertex-wise deformations of an ellipsoid and increase the mesh resolution in a coarse-to-fine manner. Such a strategy has also been adopted in Voxel2Mesh [55] to reconstruct simple human organ surfaces (e.g., liver, hippocampus) from CT/MR images. Similarly, PialNN [31] learns to deform an input inner cortical (i.e., WM) surface to a target outer (pial) surface by a sequence of deformation blocks; TopoFit [20] warps a topologically-correct template surface to fit the WM surface by a series of graph convolutional blocks; and Vox2Cortex [10] deforms a brain template surface to target cortical surfaces by leveraging combined CNNs and GNNs. Another line of research leverages ODE to parameterize the deformation of vertices as diffeomorphic flows, such as CoticalFlow methods [26, 47] and CortexODE [30]. However, all these methods may produce intersecting inner and outer cortical surfaces because they predict the inner and outer cortical surfaces either using separate DL models or using a joint DL model without explicit constraints to penalize the generation of intersecting surfaces.

## 2.2 Diffeomorphic Deformation

A diffeomorphism is theoretically differentiable and invertible and guarantees smooth and one-to-one mapping [46]. Diffeomorphic deformation has been widely used in image registration [2, 4, 5, 35, 51]. It can be generated from a velocity field $\mathbf{v}$ by integrating an ordinary differential equation (ODE) [1],

$$\frac{d\phi(\mathbf{x}, t)}{dt} = \mathbf{v}(\phi(\mathbf{x}, t), t), \text{and thus } \phi(\mathbf{x}, t) = \phi(\mathbf{x}, 0) + \int_o^t \mathbf{v}(\phi(\mathbf{x}, t), t) dt, \quad (1)$$

where $\phi(\mathbf{x}, 0) = \mathbf{x}$. The velocity field can be stationary [4] or time-varying [8]. Standard numerical integration techniques, such as the Euler method and the Runge-Kutta method [11], can be used to perform the integration. CoticalFlow methods [26, 47] and CortexODE [30] parameterize the ODE by a neural network [12]. CoticalFlow requires the deformation field to be Lipschitz at each time step to ensure a bijective mapping with Lipschitz inverse and derives a stability condition for the numeric approximation of $\phi$. As a chain of deformation modules is trained in sequential stages to predict a series of deformation fields that deform the initial mesh template to the target surface in a coarse-to-fine manner, there is a risk of generating self-intersections due to the low-resolution template in the coarsest level model and the whole process is prone to error accumulation in multiple steps. CortexODE [30] is built upon NODE [12] and operates on a topologically-corrected high-resolution initialization mesh. The sufficient condition of diffeomorphism can be satisfied if the deformation network is Lipschitz continuous and a sufficiently small step size is used in numerical approximation. Most learning-based image registration approaches [6, 28, 36] adopt stationary velocity field (SVF), and the integral (displacement field) is computed on a voxel-grid using the scaling and squaring method [3], yielding comparable registration accuracy to conventional methods while significantly improving efficiency. Our work lies at the intersection of these methods in that our method uses CNNs to parameterize multiple SVFs based on a multi-channel input and train the SVFs jointly to optimize cortical surfaces under topology-preserving and inverse-consistent transformation regularizations.

## 3 Methodology

As illustrated in Fig. 1, our framework consists of two parts: a pipeline that estimates a midthickness surface, represented both explicitly as a 3D mesh and implicitly as a 3D distance map (Sect. 3.1), and

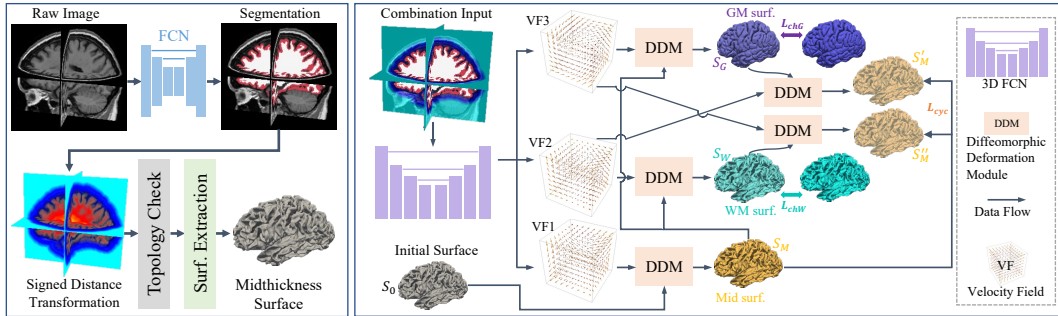

|  |  |
|:---:|:---:|
| (a) Surface initialization | (b) Cortical surface reconstruction network |

Figure 1: Overview. (a) Surface initialization. An FCN generates cortical ribbon segmentation maps from the raw image. An SDF is derived, followed by topology correction and marching cubes algorithms, and then used for initializing the midthickness surface. (b) Cortical surface reconstruction network. It takes a multi-channel input and learns three parallel velocity fields (VFs) in parallel. $VF_1$ is used to generate a deformation field for optimizing the midthickness surface $\mathcal{S}_M$. $VF_2$ (or $VF_3$) is used to deform $\mathcal{S}_M$ to the WM surface $\mathcal{S}_W$ (or pial surface $\mathcal{S}_G$) and then $VF_3$ (or $VF_2$) deforms $\mathcal{S}_W$ (or $\mathcal{S}_G$) to midthickness surface $\mathcal{S}''_M$ (or $\mathcal{S}'_M$) to constrain topology between cortical surfaces.

an end-to-end fully convolutional network (FCN) that reconstructs multiple surfaces and estimate cortical thickness simultaneously (Sect. 3.2). Loss functions are presented in (Sect. 3.3).

## 3.1 Midthickness Surface Initialization

The human cerebral cortex is a 2D sheet with an average thickness of $\sim 2.5mm$ and has a highly folded geometry with peaks (i.e., gyri) and grooves (i.e., sulci) [17]. The existing CSR methods typically use a smoothed template surface estimated from a group of subjects [10, 47], or a WM surface [30, 31] to initialize the surface reconstruction. However, our preliminary experiment shows that the closer the initial surface is to its target surface, the higher the reconstruction accuracy is (see Fig. 2(a)). Thus, we propose to extract the midthickness layer which lies halfway between the inner and outer cortical surfaces as an initialization surface, which brings three advantages: (1) Deforming a surface from the midthickness surface reduces the learning difficulty and improves accuracy by avoiding learning "large" deformations. It also strikes a balance between optimizing the inner and outer surfaces, making it less challenging. (2) The distance between the midthickness surface and the pial surface can help prevent topology errors in cases of deep sulci, as this layer still provides a clear separation between them. (3) Deforming the midthickness surface inward and outward to the inner and outer surfaces establishes a one-to-one mapping that explicitly encodes the correspondence between surfaces, facilitating coupled surface learning and improving CSR accuracy.

A key challenge is how to obtain a surface as close to the midthickness surface as possible. A straightforward method is to inflate the WM surface along the normal direction by a pre-determined distance [30]. However, this method cannot accurately estimate the midthickness surface in that the cortical thickness varies across the cortex and the normal is merely an approximation using neighboring faces. We propose a more accurate method by leveraging the cortex ribbon segmentation map (i.e., the filled interior area of WM and pial surfaces). Given an input brain MRI volume $I \in \mathbb{R}^{D_1 \times D_2 \times D_3}$, we utilize a 3D U-Net [44] to generate a WM segmentation map $M_W \in \mathbb{R}^{D_1 \times D_2 \times D_3}$ and a GM segmentation map $M_G \in \mathbb{R}^{D_1 \times D_2 \times D_3}$ (see Fig. 1(a)). The network is trained on a large-scale public neuroimaging dataset [22] by minimizing the cross-entropy loss between the prediction and ground truth which can be obtained using existing pipelines [17, 45]. Based on the predicted segmentation map $M_W$, we generate a signed distance function (SDF), $K_W \in \mathbb{R}^{D_1 \times D_2 \times D_3}$, using a distance transform algorithm: $d(v_i) = SDF(v_i)$ is the minimal Euclidean distance of voxel $v_i \in I$ to the boundary voxels. Voxels with values equal to zero represent the surface boundaries and voxels with negative or positive values encode their distances to the surface boundaries inward or outward, respectively. Similarly, we generate an SDF, $K_G$, for the pial (gray matter) surface. A new SDF can be obtained by averaging the WM and GM SDFs: $K_M = (K_G + K_W)/2$, whose 0-level defines the midthickness surface implicitly. To ensure the midthickness surface maintains a spherical topology, a fast topology check and correction algorithm [7, 30] is then applied to the implicit surface $K_M$.

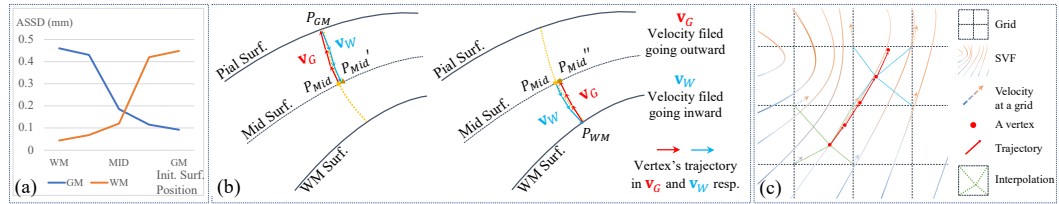

Figure 2: (a) The relationship between the initialization surface position (X-axis) and the surface reconstruction accuracy (Y-axis) using PialNN [31]. Orange and blue lines represent WM and pial surfaces reconstruction resp. (b) Illustration of a symmetric deformation trajectory and cortical thickness estimation. (c) Illustration of DDM. Each vertex is deformed by sampled velocities.

Finally, the initialization midthickness surface is extracted by the marching cubes algorithm [27] from the 0-level of $K_M$ and parameterized by a triangular mesh $\mathcal{S}_0$.

## 3.2 Coupled Reconstruction of Cortical Surfaces

Based on the formulation in Eq. 1, an accurate parameterization of the SVFs is crucial to modeling the diffeomorphic trajectory of each vertex and reconstructing the cortical surfaces. In this section, we will show how our method achieves the goals.

**Feature Extraction from a Mutli-channel Input of Brain MRI, Cortical Ribbon, and Cortical Surface.** The existing DL-based CSR studies have demonstrated that it is critical to fuse both image features extracted from the MRIs using CNNs and geometry features extracted from surface meshes using GNNs/MLPs. However, besides employing two different network architectures, these methods typically learn the image and geometry features *separately* before fusion, which does not adequately utilize image texture and surface geometric information. To better model the SVFs, we propose to learn features from a multi-channel input consisting of a brain MRI, its cortical ribbon segmentation maps, and its midthickness surface represented as an SDF: $I_{comb} = I © M_{W \oplus G} © K_M$, where © is channel-wise concatenation and $M_{W \oplus G}$ is a multiclass mask (BG = 0, WM = 0.5, GM = 1).

Such a feature learning procedure brings two key benefits: (1) Utilizing heterogeneous features enables mutual knowledge distillation. The brain MRI contains detailed texture and semantic information but may include noise and irrelevant regions far from the target surfaces. The cortical ribbon segmentation maps contain structural/semantic information about the cortical sheet and can act as an attention guide for extracting informative features around its boundaries. The SDF implicitly embeds the surface location information and relative relation between all voxels. Together, the multi-channel input provides richer and complementary information for our model to reconstruct the surfaces. (2) Only a single CNN is needed, which simplifies the model design and improves efficiency. Features can be extracted in a single forward pass for all coordinates. When scaling up the number of vertices in mesh, we can interpolate in the feature space efficiently.

**Coupled Learning of Cortical Surfaces.** The goal is to learn diffeomorphic deformations that deform the initialization midthickness surface $\mathcal{S}_0 \subset \mathbb{R}^3$ to its target WM and pial surfaces, $\mathcal{S}_W$ and $\mathcal{S}_G$. Taking into account the discrepancy between the initialization $\mathcal{S}_0$ and the *true* midthickness surfaces $\mathcal{S}_M$, our method also learns a diffeomorphic deformation to optimize the initialization midthickness surface. In total, our method learns a function to model three diffeomorphic deformations $f_\theta(I_{comb}, \mathcal{S}_0) = (\phi_M, \phi_W, \phi_G)$, using an FCN and several diffeomorphic deformation modules (DDMs). Specifically, the FCN has a similar architecture as U-Net [44], consisting of a 5-level hierarchical encoder-decoder with skip connections as shown in Fig. 1(b) (see Supplementary Materials for details). By taking the multi-channel input, the FCN learns to estimate three dense SVFs jointly, denoted by $\mathbf{v}_M$, $\mathbf{v}_W$, and $\mathbf{v}_G$. We then use $\mathbf{v}_M$ to compute $\phi_M$ that deforms $\mathcal{S}_0$ to the *true* midthickness surface $\mathcal{S}_M$, $\mathbf{v}_W$ to compute $\phi_W$ that deforms $\mathcal{S}_M$ inward to the WM surface $\mathcal{S}_W$, and $\mathbf{v}_G$ to compute $\phi_G$ that deforms $\mathcal{S}_M$ outward to the pial surface $\mathcal{S}_G$. By doing so, we establish a *one-to-one mapping* across $\mathcal{S}_W$, $\mathcal{S}_M$, and $\mathcal{S}_G$.

However, since $\phi_W$ and $\phi_G$ are computed using different SVFs, they may cause non-invertible transformation around the midthickness surface. To address this issue, we utilize the property of diffeomorphic mapping to compute a symmetric deformation trajectory of each vertex and devise a

symmetric cycle function $\mathcal{L}_{cyc}$ (Eq. 2) for training. Fig. 2(b) illustrates the deformation of a vertex $\mathbf{p}_{Mid}$ outward to $\mathbf{p}_{GM}$ using $\phi_G$ ($\mathbf{v}_G$) followed by the deformation inward to $\mathbf{p}'_{Mid}$ using $\phi_W$ ($\mathbf{v}_W$), which should be as close to $\mathbf{p}_{Mid}$ as possible. Similarly, $\mathbf{p}''_{Mid}$ should also be as close to $\mathbf{p}_{Mid}$ as possible when deformed inward by $\phi_W$ followed by the outward deformation by $\phi_G$. The symmetric cycle loss is formulated as:

$$\mathcal{L}_{cyc} = \frac{1}{N} \sum_{\mathbf{p} \in \mathcal{S}_M} \|\mathbf{p}_{\phi_W \circ \phi_G} - \mathbf{p}\|_2^2 + \|\mathbf{p}_{\phi_G \circ \phi_W} - \mathbf{p}\|_2^2, \tag{2}$$

where $\mathbf{p}_{\phi_b \circ \phi_a}$ represents deforming a vertex $\mathbf{p}$ using velocity fields $\mathbf{v}_a$ and $\mathbf{v}_b$ sequentially. It ensures simultaneous alignment of $\mathbf{p}_{\phi_W \circ \phi_G} \simeq \mathbf{p} \simeq \mathbf{p}_{\phi_G \circ \phi_W}$ for each vertex $\mathbf{p} \in \mathcal{S}_M$. Fig. 3 demonstrates the effectiveness of $\mathcal{L}_{cyc}$ in the *coupled* reconstruction of multiple surfaces in the network optimization. Moreover, we can trace the geodesic trajectory of each vertex during the mesh deformation process for estimating vertex-wise cortical thickness (Fig. 2(b)).

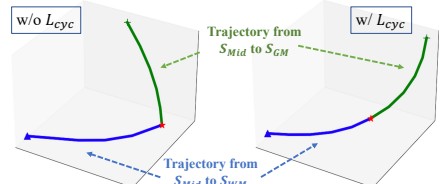

Figure 3: Example vertex deformation trajectories with and without $\mathcal{L}_{cyc}$.

**Diffeomorphic Deformation Module (DDM).** To numerically solve the ODE in Eq. 1, scaling and squaring (SS) method [3] can be applied on the image grid, followed by trilinear interpolation (Lint($\cdot$)) for vertices in continuous coordinates. However, there are three limitations: (1) Interpolation cannot guarantee the invertibility of diffeomorphic mapping; (2) Numerical errors are amplified in the two sequential steps of SS+Lint($\cdot$); (3) Computation on grid voxels including no vertices is unnecessary. Hence, we utilize the DDM to directly compute the vertex-wise integral. As shown in Fig. 2(c), we obtain the velocity vector for a vertex with coordinate $\mathbf{x}$ by interpolating its neighboring velocity vectors ($:= \overrightarrow{v}_{\mathcal{N}(\mathbf{x})}$), i.e., $\overrightarrow{v}_\mathbf{x} = \text{Lint}(\overrightarrow{v}_{\mathcal{N}(\mathbf{x})})$. The vertex then moves to a new coordinate $\overrightarrow{v}_\mathbf{x} \cdot \frac{1}{T}$, where $T$ is the total time steps. We can obtain overall deformation using this procedure in $T$ steps.

### 3.3 Loss Functions

We design multiple loss functions to optimize the geometric precision of the reconstructed surfaces and regularize the SVFs for diffeomorphic deformation.

**Mesh loss**. It aims to minimize distances of the vertices between the predicted surface meshes $\mathcal{S}_W$ (and $\mathcal{S}_G$) and their corresponding ground truth (GT) meshes $\mathcal{S}_*$ by the bidirectional Chamfer distance [26]:

$$\mathcal{L}_{chW} = \sum_{\mathbf{p} \in \mathcal{S}_W} \min_{\mathbf{p}_* \in \mathcal{S}_{W_*}} \|\mathbf{p} - \mathbf{p}_*\|_2^2 + \sum_{\mathbf{p}_* \in \mathcal{S}_{W_*}} \min_{\mathbf{p} \in \mathcal{S}_W} \|\mathbf{p}_* - \mathbf{p}\|_2^2, \tag{3}$$

where $\mathbf{p}$ and $\mathbf{p}_*$ are the coordinates of vertices on meshes. We can compute $\mathcal{L}_{chG}$ analogously. The mesh loss is $\mathcal{L}_{ch} = \mathcal{L}_{chW} + \mathcal{L}_{chG}$.

**Trajectory loss.** Starting from the midthickness surface, the trajectory length of the vertex moving to the WM and the pial surfaces should be equal. We propose to compute the mean square difference of the vertex's trajectories:

$$\mathcal{L}_{dist} = \frac{1}{N} \sum_{\mathbf{p} \in \Omega} \|L_{Mid \rightarrow GM}(\mathbf{p}) - L_{Mid \rightarrow WM}(\mathbf{p})\|_2^2, \tag{4}$$

where $L_{Mid \rightarrow GM}(\mathbf{p}) = \sum_{t=0}^{T} \Delta\phi_{G,t}(\mathbf{p})$ is the accumulated Euclidean distance (i.e., trajectory length) of $T$ steps of deformation. This term encourages the midthickness surface can be deformed to the inner and outer cortical surfaces with the same deformation path length. In other words, the midthickness surface should lie halfway between the inner and outer cortical surfaces.

**Symmetric cycle loss.** We formulate it as Eq. 2 to encourage $\phi_W$ and $\phi_G$ to be invertible.

**Symmetric similarity loss.** To optimize the midthickness surface to lie halfway between the inner and outer cortical surfaces, a magnitude difference constraint is adopted directly on the SVFs:

$$\mathcal{L}_{ss} = \|\mathbf{v}_G - \overline{\mathbf{v}}_W\|_2^2, \tag{5}$$

where $\overline{\mathbf{v}}_W$ represents reverse-directional $\mathbf{v}_W$.

**Normal consistency loss.** We also incorporate a normal consistency regularization term to promote robust learning of the surfaces and ensure their smoothness:

$$\mathcal{L}_{nc} = \sum_{e \in E, f_0 \cap f_1 = e} (1 - cos(\mathbf{n}_{f_0}, \mathbf{n}_{f_1})), \tag{6}$$

where $e$ is an edge, $f_0$ and $f_1$ are $e$'s two neighboring faces with their unit normals $\mathbf{n}_{f_0}$ and $\mathbf{n}_{f_1}$.

In summary, we combine all the losses to jointly optimize our DL model: $\mathcal{L} = \lambda_1 \mathcal{L}_{ch} + \lambda_2 \mathcal{L}_{dist} + \lambda_3 \mathcal{L}_{cyc} + \lambda_4 \mathcal{L}_{ss} + \lambda_5 \mathcal{L}_{nc}$, where $\{\lambda_i\}_{i=1,\cdots,5}$ are weights to balance the loss terms. We empirically set $\lambda_i = 1$ $(i = 1, \cdots, 4)$ and $\lambda_5 = 0.001$.

## 4  Experiments

We evaluated our method for reconstructing both white-matter (WM) and pial surfaces on two large-scale datasets, (ADNI) [22] and OASIS [33], and compared it with state-of-the-art (SOTA) DL-based CSR methods. We also tested its robustness and performed ablation analyses.

**Datasets.** The ADNI-1 [22] dataset consists of 817 subjects and we randomly split it into 654, 50, and 113 subjects for training, validation, and testing, respectively. The OASIS-1 [33] dataset consists of 413 subjects and we randomly split it into 330, 25, and 58 for training, validation, and testing, respectively. The models were trained on the training set until they reached a loss plateau on the validation set, after which their performance was evaluated on the test set. We followed pre-processing protocols in previous works [10, 13, 26, 30] for fair comparison. The T1-weighted MRI scans were aligned rigidly to the MNI152 template and clipped to the size of $192 \times 224 \times 192$ at $1mm^3$ isotropic resolution. The pseudo ground-truth of ribbon segmentation and cortical surfaces were generated using FreeSurfer v7.2.0 [17]. The intensity values of MRI scans, ribbon segmentation maps, and SDFs were normalized to $[0, 1]$ and the coordinates of the vertices were normalized to $[-1, 1]$. The WM and GM in the cortical ribbon segmentation maps were assigned values of 0.5 and 1 respectively.

**Implementation details.** Our framework was implemented in PyTorch [40] and trained on an NVIDIA P100 GPU of 16 GB memory. The 3D U-Net [44] for ribbon segmentation was trained for 200 epochs using Adam [23] optimization and achieved an average Dice index of 0.96 on the testing set. The CSR model was trained for 400 epochs using Adam ($\beta_1 = 0.9$, $\beta_2 = 0.999$, $\epsilon = 1e^{-10}$, learning rate $1e^{-4}$) to optimize the midthickness surface and reconstruct the WM and pial surfaces for each hemisphere. The surface meshes had $\sim 130K$ vertices.

**Evaluation metrics.** We utilized three distance-based metrics to measure the CSR accuracy, including Chamfer distance (CD), average symmetric surface distance (ASSD), and Hausdorff distance (HD). In particular, CD measures the mean distance between two sets of vertices [16, 52]; ASSD and HD measure the average and maximum distances between two surfaces [13, 50]. They are computed bidirectionally over $\sim 130K$ points uniformly sampled from the predicted and target surfaces. A lower distance indicates a better result. We used the 90th percentile instead of the maximum because HD is sensitive to outliers [21]. We also utilized the ratio of self-intersection faces (SIF) to measure surface quality [13, 30].

### 4.1  Comparison with Related Works

From the two categories of existing DL-based cortical surface reconstruction methods described in Section 1, we selected representative ones from each category for comparison. The experimental results are summarized in Table 2 and illustrated in Figure 4.

**Main Results & Analysis.** It is evident that our method achieved substantial improvement on both the WM and pial surface reconstruction over other approaches. Since DeepCSR [13] predicts an SDF-based implicit surface and requires post-processing to correct topology and extract a mesh, its results may contain no SIFs but were less accurate compared with the explicit CSR methods. Starting from the WM surface, PialNN [31] can achieve sub-voxel accuracy but its SIF ratio was relatively high. Vox2Cortex [30] can generate multiple surfaces from different template meshes. It employs complex CNN and GNN models to model the deformation for each vertex but has no diffeomorphism guarantee. The promising results of [26, 30, 47] indicated that using neural networks to parameterize the ODE can facilitate the diffeomorphic deformation, yielding better CSR accuracy. Our method

Table 2: Quantitative analysis of cortical surface reconstruction on geometric accuracy and surface quality. The Chamfer distance (CD), average symmetric surface distance (ASSD), Hausdorff distance (HD), and the ratio of the self-intersecting faces (SIF) were measured for WM and pial surfaces on the two datasets. The mean value and standard deviation are reported. The best ones are in bold.

| | Method | L-Pial Surface | | | | L-WM Surface | | | |
|---|---|---|---|---|---|---|---|---|---|
| | | CD ($mm$) | ASSD ($mm$) | HD ($mm$) | SIF (%) | CD ($mm$) | ASSD ($mm$) | HD ($mm$) | SIF (%) |
| ADNI | DeepCSR [13] | 0.945±0.078 | 0.593±0.065 | 1.149±0.203 | \ | 0.938±0.076 | 0.587±0.064 | 1.137±0.193 | \ |
| | PialNN [31] | 0.621±0.035 | 0.465±0.044 | 1.002±0.106 | 0.137±0.093 | \ | \ | \ | \ |
| | CorticalFlow [26] | 0.691±0.043 | 0.497±0.049 | 1.106±0.115 | 0.149±0.087 | 0.641±0.037 | 0.465±0.042 | 0.996±0.100 | 0.108±0.073 |
| | CorticalFlow++ [47] | 0.545±0.036 | 0.410±0.033 | 0.886±0.069 | 0.098±0.067 | 0.544±0.034 | 0.401±0.030 | 0.878±0.066 | 0.069±0.042 |
| | cortexODE [30] | 0.476±0.017 | 0.214±0.020 | 0.455±0.058 | **0.022**±0.012 | 0.458±0.016 | 0.192±0.015 | 0.436±0.014 | 0.015±0.011 |
| | Vox2Cortex [10] | 0.582±0.028 | 0.370±0.025 | 0.746±0.057 | 0.059±0.039 | 0.577±0.027 | 0.353±0.022 | 0.722±0.055 | 0.043±0.023 |
| | Ours | **0.410**±0.016 | **0.136**±0.012 | **0.293**±0.026 | 0.035±0.021 | **0.213**±0.008 | **0.071**±0.005 | **0.155**±0.012 | **0.007**±0.010 |
| OASIS | DeepCSR [13] | 0.986±0.085 | 0.617±0.070 | 1.331±0.212 | \ | 0.975±0.081 | 0.594±0.067 | 1.151±0.197 | \ |
| | PialNN [31] | 0.635±0.032 | 0.460±0.038 | 0.993±0.082 | 0.141±0.096 | \ | \ | \ | \ |
| | CorticalFlow [26] | 0.687±0.040 | 0.495±0.047 | 1.082±0.110 | 0.147±0.086 | 0.637±0.035 | 0.462±0.040 | 0.992±0.097 | 0.101±0.070 |
| | CorticalFlow++ [47] | 0.531±0.035 | 0.399±0.030 | 0.812±0.057 | 0.088±0.045 | 0.529±0.033 | 0.398±0.030 | 0.810±0.055 | 0.086±0.042 |
| | cortexODE [30] | 0.481±0.019 | 0.218±0.021 | 0.461±0.062 | **0.026**±0.015 | 0.463±0.018 | 0.207±0.017 | 0.435±0.015 | 0.018±0.010 |
| | Vox2Cortex [10] | 0.588±0.032 | 0.381±0.030 | 0.750±0.063 | 0.061±0.037 | 0.581±0.028 | 0.375±0.027 | 0.731±0.059 | 0.046±0.027 |
| | Ours | **0.442**±0.014 | **0.161**±0.012 | **0.348**±0.025 | 0.037±0.023 | **0.218**±0.007 | **0.073**±0.006 | **0.159**±0.013 | **0.008**±0.011 |

achieved the overall best performance due to its explicit regularizations on the deformation trajectory of vertices and its better initialization surface. On the ADNI dataset, our method achieved ∼48.8% improvement in mean ASSD (of WM and pial surfaces) compared to the second best CortexODE (i.e., $0.104mm$ *v.s* $0.203mm$) with competitive self-intersection ratio (average ∼0.021%). On the OASIS dataset, our method achieved similar performance improvement. More quantitative results on the WM surface are reported in the Supplementary Materials. As shown in Fig. 4, the CSR results obtained by our method had uniformly smaller errors across the whole surfaces.

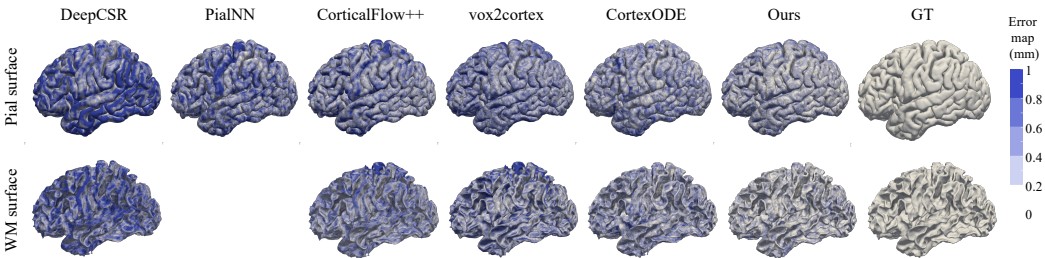

Figure 4: Visualization of the reconstructed surfaces with distance compared to ground truth.

**Runtime Analysis.** It took ∼0.8$s$ for our method to obtain a ribbon segmentation map and another ∼2$s$ for the topology correction and surface initialization. For the surface reconstruction, the inference time for simultaneously reconstructing *three* (i.e., WM, pial, and midthickness) surfaces was ∼1.5$s$. In comparison, the SOTA explicit CSR method CortexODE [30] needed ∼2$s$ to reconstruct two surfaces *sequentially*. Overall, our method is computationally as efficient as the SOTA alternatives.

### 4.2 Ablation Studies

**Input.** Our method takes as input a multi-channel 3D images $I_{comb}$ and an initialization midthickness surface $S_0$. We conducted two experiments to analyze the influence of the input components on the CSR performance (Table 3 Top). *First*, by gradually removing components from $I_{comb}$, we observed a significant drop in accuracy, indicating the contribution of both the SDF and segmentation maps to the final results. *Second*, we investigated the impact of the fine structure of the initialization surface by applying Laplacian smoothing on $S_0$ to generate oversmoothed initialization surfaces. The results revealed that the accuracy of the model decreased with the increasing of smoothing steps which resulted in a coarser initialization surface $S_0$.

**Loss functions.** We evaluated the contribution of different losses of our method to the surface reconstruction performance in terms of both accuracy (CD, ASSD, HD) and topological correctness (SIF), with the results summarized in Table 3 Middle. Through the ablation studies, we observed

Table 3: Ablation studies with the surface reconstruction results quantified in terms of CD, ASSD, HD, and SIF. The setting I0, S0, and $T = 5$ all refer to our complete setting (cf. Table 2). Top: The impact of the input image and initialization surface. Middle: The impact of the loss functions. Bottom: The impact of the deformation steps in DDM.

| Setting | Input | | | Initial mesh | L-Pial Surface | | | L-WM Surface | | |
|---|---|---|---|---|---|---|---|---|---|---|
| | I | SDF | Seg | # of Lap. Sm. | CD $(mm)$ | ASSD $(mm)$ | HD $(mm)$ | CD $(mm)$ | ASSD $(mm)$ | HD $(mm)$ |
| I0 | ✓ | ✓ | ✓ | 0 | 0.410±0.016 | 0.136±0.012 | 0.293±0.026 | 0.213±0.008 | 0.071±0.005 | 0.155±0.012 |
| I1 | ✓ | ✓ | | 0 | 0.426±0.017 | 0.167±0.017 | 0.358±0.038 | 0.222±0.011 | 0.075±0.006 | 0.164±0.013 |
| I2 | ✓ | | | | 0.453±0.021 | 0.201±0.026 | 0.438±0.074 | 0.250±0.013 | 0.085±0.008 | 0.184±0.016 |
| M1 | ✓ | ✓ | ✓ | 10 | 0.416±0.016 | 0.147±0.013 | 0.315±0.028 | 0.225±0.010 | 0.084±0.007 | 0.184±0.015 |
| M2 | ✓ | ✓ | ✓ | 20 | 0.429±0.018 | 0.163±0.017 | 0.361±0.040 | 0.235±0.012 | 0.091±0.009 | 0.190±0.017 |

| Setting | Loss | | | | | L-Pial Surface | | | | L-WM Surface | | | |
|---|---|---|---|---|---|---|---|---|---|---|---|---|---|
| | $\mathcal{L}_{CH}$ | $\mathcal{L}_{dist}$ | $\mathcal{L}_{cyc}$ | $\mathcal{L}_{ss}$ | $\mathcal{L}_{nc}$ | CD $(mm)$ | ASSD $(mm)$ | HD $(mm)$ | SIF (%) | CD $(mm)$ | ASSD $(mm)$ | HD $(mm)$ | SIF (%) |
| S0 | ✓ | ✓ | ✓ | ✓ | ✓ | 0.410±0.016 | 0.136±0.012 | 0.293±0.026 | 0.035±0.021 | 0.213±0.008 | 0.071±0.005 | 0.155±0.012 | 0.007±0.010 |
| S1 | ✓ | ✓ | ✓ | ✓ | | 0.412±0.016 | 0.138±0.012 | 0.299±0.026 | 0.036±0.021 | 0.213±0.010 | 0.073±0.006 | 0.158±0.013 | 0.008±0.010 |
| S2 | ✓ | ✓ | ✓ | | | 0.412±0.016 | 0.139±0.012 | 0.302±0.027 | 0.037±0.022 | 0.211±0.009 | 0.073±0.007 | 0.158±0.013 | 0.008±0.011 |
| S3 | ✓ | ✓ | | | | 0.409±0.016 | 0.135±0.012 | 0.300±0.027 | 0.275±0.100 | 0.209±0.009 | 0.073±0.007 | 0.156±0.013 | 0.008±0.011 |
| S4 | ✓ | | | | | 0.404±0.015 | 0.129±0.011 | 0.278±0.024 | 2.522±0.791 | 0.203±0.009 | 0.069±0.006 | 0.153±0.013 | 0.009±0.012 |

| Number of steps ($T$) in DDM | Surface | | | | Surface | | | |
|---|---|---|---|---|---|---|---|---|
| | CD $(mm)$ | ASSD $(mm)$ | HD $(mm)$ | SIF(%) | CD $(mm)$ | ASSD $(mm)$ | HD $(mm)$ | SIF(%) |
| 5 | 0.410±0.016 | 0.136±0.012 | 0.293±0.026 | 0.035±0.021 | 0.213±0.008 | 0.071±0.005 | 0.155±0.012 | 0.007±0.010 |
| 4 | 0.411±0.015 | 0.137±0.012 | 0.295±0.026 | 0.038±0.022 | 0.208±0.009 | 0.072±0.006 | 0.157±0.012 | 0.008±0.011 |
| 3 | 0.411±0.016 | 0.138±0.012 | 0.297±0.026 | 0.044±0.025 | 0.205±0.009 | 0.072±0.006 | 0.158±0.013 | 0.009±0.012 |
| 2 | 0.415±0.016 | 0.143±0.013 | 0.309±0.028 | 0.432±0.110 | 0.204±0.008 | 0.075±0.005 | 0.166±0.012 | 0.011±0.013 |

that each component played its own role in a complementary way. The first row (referred to as S0) corresponds to our complete network setting, while the last row (S4) represents using Chamfer distance alone. The results of setting S4 indicated that the model generated surfaces well matched to the ground truth data at the cost of high topological errors, particularly in highly curved regions, reflected by the results that the SIF ratio was worse on the pial surface than on the WM surface. Enforcing equality of the trajectories from the midthickness surface to the WM and pial surfaces (S3, $\mathcal{L}_{dist}$) helped optimize the midthickness surface, thereby preventing deformation in an arbitrary direction and reducing self-intersection. However, the geometric accuracy slightly decreased, which might be caused by the difficulty in accessing highly curved regions or deep sulci under such strong topology constraints. The proposed symmetric cycle loss (S2, $\mathcal{L}_{cyc}$) promoted the invertibility of deformations, yielding a significant reduction of self-intersections on the meshes since our method jointly reconstructs both the inner and outer surfaces by deforming the midthickness surface inward and outward with two VFs. Such invertibility also facilitates accurate estimation of the cortical thickness from the trajectory, as illustrated by a sample vertex deformation trajectory in Fig. 2 obtained with settings of S2 and S3. Moreover, the inclusion of regularization terms on the smoothness of SVFs (S1, $L_{ss}$) and surfaces (S0, $L_{nc}$) contributed to enhancement in surface quality. Overall, our proposed method struck a balance between geometric accuracy and topology quality.

**Deformation steps in DDM.** Table 3 Bottom shows the evaluation results of different numbers of deformation steps ($T$) in DDM. As $T$ increased, the performance first improved and then saturated, indicating that five steps were sufficient to deform the midthickness surface to the inner or outer surfaces.

### 4.3 Reproducibility

We carried out two experiments on two datasets: a paired ADNI$_{1.5\&3T}$ dataset [22] consisting of 1.5T and 3T MRIs of the same subjects, and the Test-Retest dataset [32] comprising 40 MRIs collected within a short period for each of the 3 subjects. In these scenarios, the cortical surfaces of the same subject should be nearly identical. Following the experimental setup outlined in [10, 13, 30], we utilized the iterative closest-point algorithm (ICP) to align image pairs and computed the geometric distance between surfaces. The results for the left WM surfaces are presented in Table 4 (more in Supplementary Materials),

Table 4: Reproducibility analysis.

| | Method | L-WM Surface | | |
|---|---|---|---|---|
| | | CD $(mm)$ | ASSD $(mm)$ | HD $(mm)$ |
| ADNI-pair | Ours | **0.520**±0.053 | **0.337**±0.058 | **0.738**±0.151 |
| | CortexODE | 0.521±0.056 | 0.340±0.060 | 0.741±0.154 |
| | DeepCSR | 0.618±0.103 | 0.397±0.080 | 0.823±0.211 |
| | FreeSurfer | 0.556±0.049 | 0.364±0.054 | 0.764±0.118 |
| TRT | Ours | **0.451**±0.019 | **0.235**±0.030 | **0.492**±0.059 |
| | CortexODE | 0.457±0.021 | 0.238±0.031 | 0.504±0.071 |
| | DeepCSR | 0.505±0.047 | 0.297±0.053 | 0.610±0.100 |
| | FreeSurfer | 0.476±0.015 | 0.253±0.022 | 0.519±0.048 |

demonstrating that our method obtained superior reproducibility compared with FreeSurfer and was comparable to the SOTA DL methods.

### 4.4 Cortical Thickness

In contrast to the alternative methods that rely on the ICP algorithm for registering WM and pial surfaces prior to calculating the Euclidean distance [10], our proposed method directly provides vertex-wise cortical thickness estimation. To validate the cortical thickness estimation, we compared our method with FreeSurfer for estimating the cortical thickness. We identified 200 subjects from the ADNI-2GO [22] dataset (100 are diagnosed with Alzheimer's disease and 100 are normal controls) and computed the average cortical thickness across 35 cortical regions based on a surface parcellation provided by FreeSurfer [17]. Fig. 5 shows the correlation between ours and FreeSurfer's results, showcasing the effectiveness of our proposed framework in accurately capturing cortical thickness.

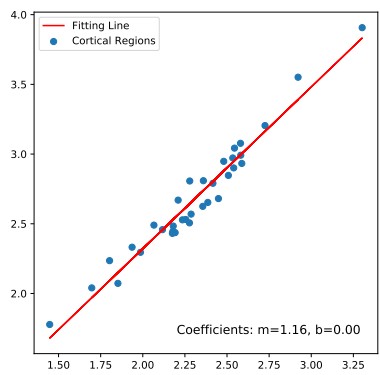

Figure 5: Correlation between prediction (Y-axis) and FreeSurfer's thickness (X-axis) on 35 cortical regions ($mm$).

## 5 Limitations and Future Directions

Despite achieving improved CSR accuracy and a low SIF ratio, our method can be further improved by adopting post-processing methods and new loss functions in order to minimize the SIF ratio and improve surface quality. While focusing on cortical thickness estimation in this paper, we recognize the value of incorporating other cortical attributes like surface area and sulci depth into CSR and analysis tasks. These attributes could serve as complementary constraints, enhancing overall performance. It should be noted that further analysis is merited to thoroughly evaluate the proposed method on a large cohort of subjects (e.g., subjects in different stages of AD) although we have demonstrated the correlation between the estimated cortical thickness and that of FreeSurfer on a balanced dataset.

## 6 Conclusion

We introduce a new DL framework for cortical surface reconstruction by generating a midthickness surface to initialize a coupled reconstruction of both the WM and pial surfaces. Specifically, the midthickness surface is estimated from a 3D distance map from each MRI by generating a cortical ribbon segmentation map that encodes structural information of the cerebral cortex. The estimated midthickness surface is represented as a triangular mesh with spherical topology, and the mesh is optimized to lie at the center of the inner and outer cortical surfaces and deformed to the inner and outer cortical surfaces by three diffeomorphic flows that are learned jointly with CNNs optimized with a multi-channel input consisting of the brain MRI, the 3D distance map of midthckness surface, and the cortical ribbon segmentation map. Our proposed symmetric cycle loss helps learn diffeomorphic deformation and the numerical solution of DDM improves CSR accuracy and computation efficiency. Experiments on two large-scale neuroimage datasets have demonstrated the superior performance of our method. Moreover, our method generates an estimation of cortical thickness, facilitating statistical analyses of brain atrophy.

## 7 Acknowledgements

This work was supported in part by the NIH grants AG066650 and EB022573.

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
