# Supplementary Materials for Coupled Reconstruction of Cortical Surfaces by Diffeomorphic Mesh Deformation

**Hao Zheng    Hongming Li    Yong Fan**
University of Pennsylvania
hzheng1@upenn.edu, {hongming.li,yong.fan}@pennmedicine.upenn.edu

## 1  Implementation Details

### 1.1  Cortical Ribbon Segmentation Network Architecture

Fig. 1 shows the network architecture of our cortical ribbon segmentation model, which consists of a 5-level hierarchical encoder-decoder with skip connections. This network takes a 3D brain MRI as its input and produces a cortex ribbon segmentation map. It is worth noting that the white matter (WM) segmentation refers to the interior of the WM surface, which contains cortical WM, deep gray matter, ventricle, hippocampus, and other tissues within the surface. Similarly, grey matter (GM) segmentation refers to the interior of the pial surface. The output map consists of 5 classes, namely left hemisphere WM and GM, right hemisphere WM and GM, and background. In the encoder, at each level, we apply a $3 \times 3 \times 3$ convolutional layer with a stride of 2 to downsample the features. In the decoder, upsampling is applied to the features at each scale by $2\times$, and the features are concatenated with their counterparts of the encoder via a skip connection, in conjunction with feature fusion using a $3 \times 3 \times 3$ convolutional layer with a stride of 1. From the input, features are learned with a $3 \times 3 \times 3$ convolutional layer with a stride of 1. The output is generated by three *consecutive* convolutional layers prior to the final prediction. Each convolutional layer is followed by a leaky ReLu activation function, except the last one where a Softmax function is adopted before computing cross-entropy loss with the ground truth.

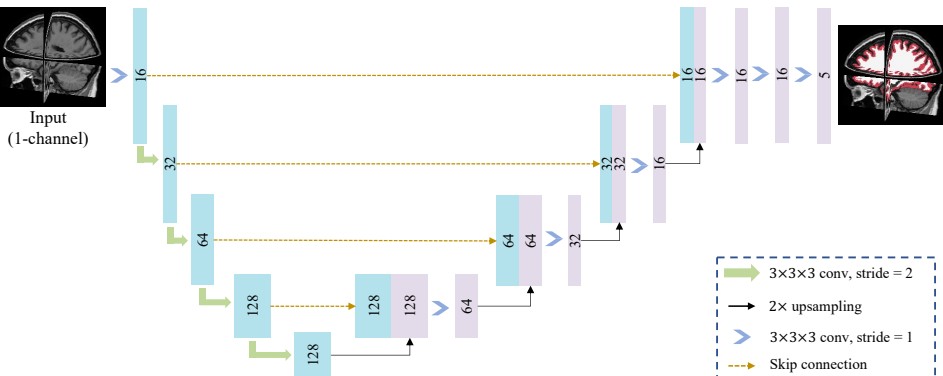

Figure 1: UNet architecture for the ribbon segmentation. The output, i.e., the cortical ribbon map is overlaid on the input image for illustration

37th Conference on Neural Information Processing Systems (NeurIPS 2023).

## 1.2 Cortical Surface Reconstruction Network Architecture and Training details

As shown in Fig. 2, our cortical surface reconstruction (CSR) network has five scales. To save computation memory, we directly employ a $3 \times 3 \times 3$ convolution with a stride of 2 to downsample the input image and do not conduct complex feature fusion via skip connection in the decoding path of this scale. To enhance the velocity fields (VFs) accuracy, we utilize $2 \times 2 \times 2$ deconvolutions with a stride of 2 in the decoding path, instead of $2\times$ trilinear upsampling. At the output stage, we employ three parallel $3 \times 3 \times 3$ convolution layers to generate VFs for WM, midthickness, and pial surfaces, respectively. ReLu activation functions are applied after each convolutional layer, except for the three parallel layers where Softsign functions are adopted. The VFs are then used to compute diffeomorphic deformations using our proposed DDM (see Fig. 2(c) in our main paper).

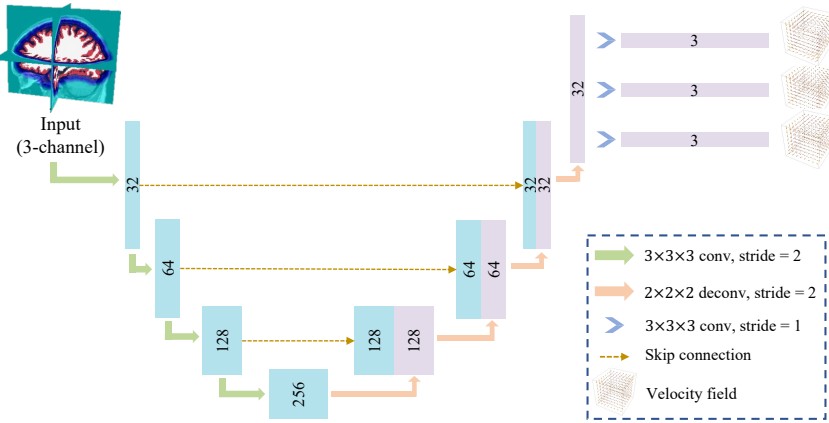

Figure 2: UNet architecture for cortical surface reconstruction.

## 1.3 Detailed Demographic Information for Datasets in Experiments

Our method has been evaluated on MRI brain scans of subjects aged 18 to 96 from multiple datasets, including ADNI-1, ADNI-2&GO, OASIS-1, and a Test-retest dataset.

The ADNI-1 dataset[1] consisted of 817 T1-weighted (T1w) brain MRIs from subjects aged 55 to 90, including normal controls (NC), subjects with mild cognitive impairment (MCI), and patients with Alzheimer's disease (AD), the ADNI-Go&2 dataset[1] consisted of 200 T1w scans from subjects aged 60 to 87, including 100 NCs and 100 ADs, the OASIS-1 dataset[2] consisted of 413 T1w scans of subjects aged 18 to 96 years, including NCs and ADs, and the Test-retest dataset[3] consisted of 120 T1w scans of three subjects aged 26 to 31. More details of the datasets can be found on their official websites.

## 1.4 Dataset Preprocessing

We preprocessed all the MRIs with the same protocols as following: Based on the standard processing protocol in FreeSurfer V7.2.0 [3], the original images were conformed and normalized (saved as `orig.mgz`), affinely registered to the MNI152 template [1] using the NiftyReg toolbox [8]. The respective ribbon segmentation maps, SDFs, and pseudo-ground-truth surfaces were also transformed using the computed transformation.

## 2 Analysis of cortical thickness estimation on AD and NC patients

We identified 200 subjects ($n_{AD} = n_{NC} = 100$) from the ADNI-2GO [4] dataset to conduct the analysis. Fig. 3 shows the details of average cortical thickness in 35 cortical regions estimated by FreeSurfer and our method. The high similarity between the results generated by our method

---

[1]ADNI website `https://adni.loni.usc.edu/`

[2]OASIS website `https://www.oasis-brains.org/`

[3]Test-retest dataset: `http://dx.doi.org/10.6084/m9.figshare.929651`

and FreeSurfer demonstrates the efficacy of our method in accurately capturing cortical thickness, indicating that our method holds promise for quantifying cortical thickness.

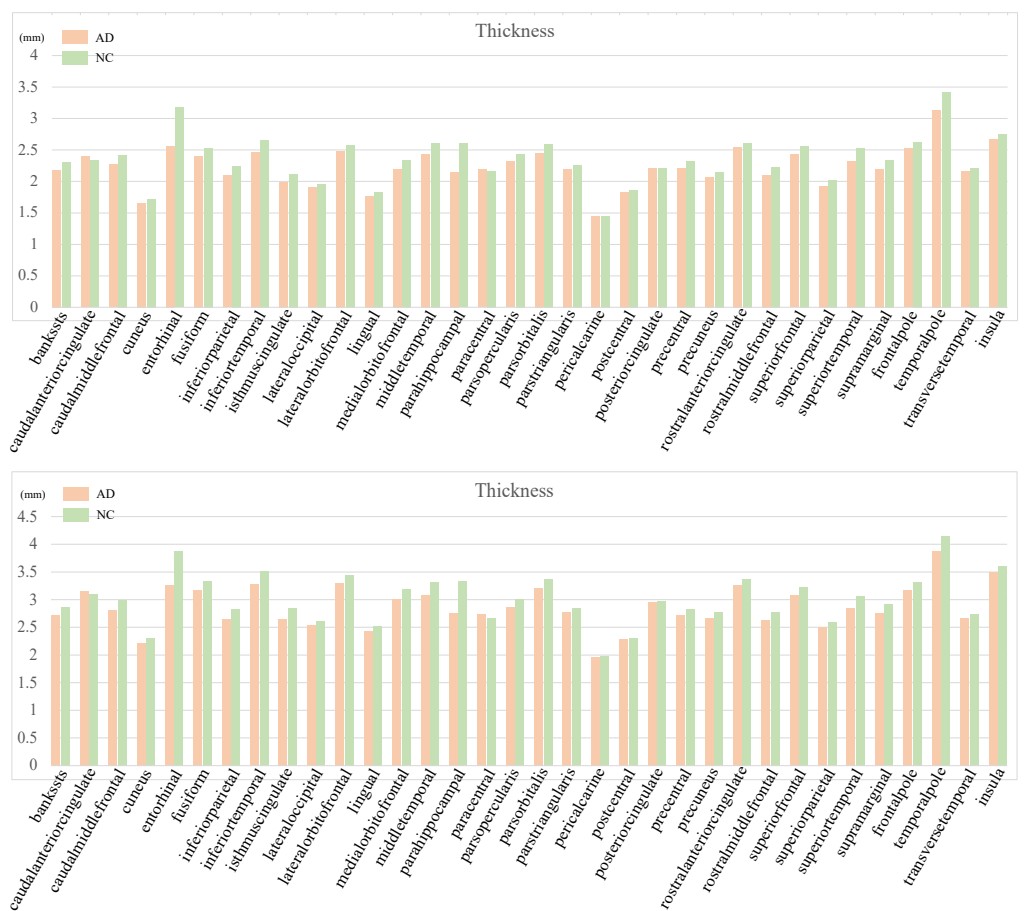

Figure 3: Average thickness in 35 cortical regions on subjects diagnosed with Alzheimer's disease (AD) and normal control (NC). Top: FreeSurfer; Bottom: Ours.

# 3 More Experimental Results

## 3.1 Quantitative Results on Right Hemisphere

Quantitative comparison results of our method and state-of-the-art (SOTA) alternatives on the *right hemisphere* are summarized in Table 1 as a supplement to Table 2 in the main paper.

## 3.2 Reproducibility Analysis on Pial Surface

Quantitative reproducibility analysis on the *pial surface* is summarized in Table 3 as a supplement to Table 4 in the main paper.

## 3.3 Impact of Spatial Resolution of SDF and Initialization Surface Granularity

The standard brain MRI image resolution is $256^3$. As a proof of concept, we applied Marching Cubes on images at a resolution of $256^3$ resolution and upsampled WM SDF at a resolution of $512^3$. The reconstructed WM surfaces' mean ASSD was reduced from $0.265mm$ to $0.254mm$ but the computation time increased from $4s$ to $5min$. A similar trend was observed on the pial surfaces with a lower geometric accuracy. We also increased the mesh granularity by mesh subdivision ($\sim 1s$ for a mesh with $130K$ vertices), but the ASSD barely changed ($0.256mm$). However, reducing

Table 1: Quantitative analysis of cortical surface reconstruction on geometric accuracy and surface quality. The Chamfer distance (CD), average symmetric surface distance (ASSD), Hausdorff distance (HD), and the ratio of the self-intersecting faces (SIF) were measured for WM and pial surfaces on the two datasets. The mean value and standard deviation are reported. The best ones are in bold.

| | Method | R-Pial Surface | | | | R-WM Surface | | | |
|---|---|---|---|---|---|---|---|---|---|
| | | CD $(mm)$ | ASSD $(mm)$ | HD $(mm)$ | SIF $(\%)$ | CD $(mm)$ | ASSD $(mm)$ | HD $(mm)$ | SIF $(\%)$ |
| ADNI | DeepCSR [2] | 0.948±0.080 | 0.597±0.068 | 1.154±0.207 | \ | 0.942±0.077 | 0.589±0.065 | 1.140±0.195 | \ |
| | PialNN [7] | 0.625±0.037 | 0.468±0.046 | 1.005±0.107 | 0.138±0.090 | \ | \ | \ | \ |
| | CorticalFlow [5] | 0.699±0.042 | 0.503±0.050 | 1.113±0.127 | 0.152±0.091 | 0.648±0.036 | 0.481±0.048 | 1.044±0.110 | 0.123±0.080 |
| | CorticalFlow++ [9] | 0.550±0.038 | 0.413±0.034 | 0.891±0.071 | 0.101±0.069 | 0.548±0.035 | 0.403±0.032 | 0.883±0.068 | 0.071±0.042 |
| | cortexODE [6] | 0.482±0.019 | 0.220±0.022 | 0.461±0.060 | **0.033**±0.017 | 0.470±0.020 | 0.207±0.019 | 0.444±0.018 | 0.023±0.016 |
| | Vox2Cortex [1] | 0.593±0.032 | 0.382±0.029 | 0.755±0.061 | 0.071±0.045 | 0.588±0.029 | 0.363±0.024 | 0.741±0.057 | 0.059±0.035 |
| | Ours | **0.419**±0.018 | **0.140**±0.015 | **0.299**±0.030 | 0.040±0.027 | **0.215**±0.008 | **0.073**±0.006 | **0.160**±0.015 | **0.008**±0.011 |
| OASIS | DeepCSR [2] | 0.989±0.086 | 0.619±0.071 | 1.336±0.215 | \ | 0.980±0.082 | 0.601±0.069 | 1.175±0.202 | \ |
| | PialNN [7] | 0.636±0.033 | 0.461±0.038 | 0.995±0.083 | 0.150±0.101 | \ | \ | \ | \ |
| | CorticalFlow [5] | 0.691±0.042 | 0.499±0.049 | 1.087±0.112 | 0.154±0.093 | 0.656±0.037 | 0.466±0.042 | 0.998±0.099 | 0.113±0.075 |
| | CorticalFlow++ [9] | 0.540±0.037 | 0.405±0.032 | 0.834±0.060 | 0.095±0.052 | 0.536±0.035 | 0.402±0.031 | 0.830±0.058 | 0.088±0.049 |
| | cortexODE [6] | 0.497±0.023 | 0.225±0.024 | 0.473±0.065 | **0.038**±0.027 | 0.481±0.021 | 0.214±0.021 | 0.450±0.022 | 0.025±0.019 |
| | Vox2Cortex [1] | 0.598±0.033 | 0.386±0.031 | 0.761±0.064 | 0.072±0.040 | 0.592±0.031 | 0.379±0.028 | 0.752±0.061 | 0.061±0.037 |
| | Ours | **0.448**±0.015 | **0.168**±0.014 | **0.353**±0.027 | 0.041±0.032 | **0.221**±0.008 | **0.080**±0.009 | **0.166**±0.016 | **0.011**±0.015 |

Table 2: Reproducibility analysis.

| | Method | L-Pial Surface | | |
|---|---|---|---|---|
| | | CD $(mm)$ | ASSD $(mm)$ | HD $(mm)$ |
| ADNI-pair | Ours | **0.519**±0.049 | **0.334**±0.057 | **0.734**±0.146 |
| | CortexODE | 0.521±0.049 | 0.339±0.055 | 0.740±0.150 |
| | DeepCSR | 0.626±0.118 | 0.412±0.091 | 0.902±0.238 |
| | FreeSurfer | 0.554±0.056 | 0.339±0.069 | 0.712±0.150 |
| TRT | Ours | **0.493**±0.024 | 0.276±0.036 | **0.573**±0.070 |
| | CortexODE | 0.506±0.029 | **0.272**±0.034 | 0.581±0.079 |
| | DeepCSR | 0.560±0.055 | 0.341±0.060 | 0.668±0.118 |
| | FreeSurfer | 0.526±0.021 | 0.283±0.032 | 0.595±0.068 |

the resolution substantially increased the reconstruction error (especially for the pial surface). In summary, simply applying Marching Cubes to SDF at a higher resolution did bring improvement in terms of geometric accuracy but incurred substantially heavier computational costs. Nevertheless, it may degrade the surface reconstruction significantly if the surface initialization is obtained based on SDFs at a lower resolution.

Following previous works, we used mesh with $\sim 130K$ vertices in our experiments. To test the impact of the number of vertices on the downstream tasks, we simplified the meshes by merging neighboring faces ($\sim 60K$), subdivided the mesh by dividing each face into four faces ($\sim 520K$), and utilized them to train different networks. We randomly chose a subset from ADNI-1 (300, 20, and 80 for training, validation, and test, respectively) to conduct the experiments. The ASSD results are as follows:

Table 3: Reproducibility analysis.

| | $60K$ | $130K$ | $520K$ |
|---|---|---|---|
| L-WM | 0.122 | 0.074 | 0.067 |
| L-Pial | 0.201 | 0.156 | 0.154 |