# OpenReview forum: "Coupled Reconstruction of Cortical Surfaces by Diffeomorphic Mesh Deformation"
_NeurIPS.cc/2023/Conference — NeurIPS 2023 poster_

### Official Review · Reviewer_vxoq · 2023-07-04

**Soundness:** 3 good
**Presentation:** 2 fair
**Contribution:** 2 fair
**Rating:** 7
**Confidence:** 4

**Summary:**

This paper is solving a solid practical problem, handling the cortical thickness estimation problem that was historically ignored. Moreover, it improves the quality of surface reconstruction. Authors observed that deforming an over-smoothed template results in lower accuracy if the template is relatively further away from the target. Therefore, reconstructing the midthickness layer could help improve the accuracy.

**Strengths:**

(1) The problem to be solved is well-motivated, limitations of existing methods are analyzed.

(2) Figures are nicely made.

(3) Storytelling is good.


**Weaknesses:**

(1) Rather than integrating all the equations within the text, it would enhance clarity if you present them as numbered equations. Furthermore, providing explanations for each variable immediately after equations would help.

(2) Paragraphs are too long. Most subsections are single-paragraphed, and all contents are presented within that one paragraph.


**Questions:**

(1) Can you detail Figure1, so that it’s more informative? For example, the variables in the equations could appear in the figure at corresponding locations.

(2) Can you split into shorter paragraphs, where each paragraph conveys one observation / insight?


**Limitations:**

Yes

---

> ### Author Rebuttal · Authors · 2023-08-08
>
> Thanks for your comments and suggestions. We will revise our paper accordingly.
>
> **C1: Rather than integrating all the equations within the text, it will enhance clarity if you present them as numbered equations. Furthermore, providing explanations for each variable immediately after equations would help.**
>
> **A1**: Due to the space limit, we put equations in text in Section. 3.3. Camera-ready version allows one additional content page, we can present them as numbered equation and provide explanation accordingly if our paper gets accepted.
>
>
> **C2: Paragraphs are too long. Most subsections are single-paragraphed, and all contents are presented within that one paragraph. Can you split into shorter paragraphs, where each paragraph conveys one observation/insight?**
>
> **A2**: Thanks. If more space allowed, we would definitely split long paragraphs into shorter ones.
>
>
> **C3: Can you detail Figure1, so that it’s more informative? For example, the variables in the equations could appear in the figure at corresponding locations.**
>
> **A3**: We will add more description in the caption and notation in the figure. Note that there was an error in the data flow in the main paper and we rectified it in the supplementary material (Fig. 1). We will update the Fig. 1 in the main paper.
>
> Fig. 1(a) illustrates the midthickness initialization process. An FCN utilizes raw images to generate white matter (WM) and gray matter (GM) segmentation maps. From these, a signed distance function (SDF) is derived, followed by topology correction and Marching Cubes algorithms, and then used for initializing the midthickness surface.
>
> Fig. 1(b) outlines the cortical thickness reconstruction network. It takes a composite input of raw images, SDF, and segmentation maps, simultaneously learning three parallel velocity fields (VFs): $\symbf{v}_M$, $\symbf{v}_W$, and $\symbf{v}_G$. The $\symbf{v}_M$ is used to generate deformation field for optimizing the midthickness surface $S_M$. The $\symbf{v}_M$ (or $\symbf{v}_G$) is used to deform $S_M$ to the WM surface $S_W$ (or pial surface $S_G$) and then deform $S_W$ (or $S_G$) to midthickness surface $S_M'$ (or $S_M''$) to constrain topology between WM and pial surfaces.

---

> > ### Comment · Reviewer_vxoq · 2023-08-13
> >
> > The rebuttal addressed all my concerns. I recommend the acceptance of this paper.

---

### Official Review · Reviewer_DGUA · 2023-07-06

**Soundness:** 2 fair
**Presentation:** 3 good
**Contribution:** 2 fair
**Rating:** 4
**Confidence:** 5

**Summary:**

This paper proposes a method for reconstructing cortical surface (CSR) (including white matter (WM) and gray matter (GM)) from MRI images. The method first estimates the “midthickness” surface, which is the surface between WM and GM, and learn jointly diffeomorphic flows to deform it to WM and GM, respectively. The proposed model is then evaluated on 2 brain dataset including ADNI and OASIS and compare with other CSR methods.


**Strengths:**

- The submission is well-written, the authors adequately demonstrate their ideas in the paper.

- In general, the proposed method provides a good quantitative result compared to other CSR methods.


**Weaknesses:**

- While I appreciate the reported improvement of the proposed method, I do not see a significant novel contribution of this work. Specifically, the idea of using Neural ODE to deform from the initial surface to target surface is widely explored in this task (CortexODE, CorticalFlow). The losses that the authors employed are also commonly applied in this task. Finally, the technical detail proposed in the paper does not seem to have a wide applicability in other areas.



**Questions:**

- In terms of qualitative results, I check the visualization of every method that the paper compares with, and none of them has equally distributed errors across the surface as illustrated in the paper (Fig 4). I wonder how the visualization is constructed to demonstrate your result.
- The paper seems to be an improvement work of CortexODE. Instead of using the initial surface that is extracted from WM segmentation, the authors extract the mid surface, which is in between the inner surface and the outer surface as the initial surface. Regarding loss functions, I wonder why the SIF of the L-Pial surface when using only Chamfer loss is so bad (2.522), but SIF of the L-WM surface is still good (0.009). Furthermore, it seems to me that adding auxiliary losses other than Chamfer loss makes the model even worse (Table 3b). Specifically, when using only Chamfer loss (S4), the model outperforms the complete network setting loss (S0) in almost every metric. Instead of using only Chamfer loss, why do the authors propose to use multiple losses but get worse performance?

**Limitations:**

The authors did not acknowledge any potential negative societal impact of their work.

---

> ### Author Rebuttal · Authors · 2023-08-09
>
> Thanks for your comments and suggestions.
>
> **C1: Novelty\
> C1-1: Neural ODE is widely used for deformation in this task (CortexODE, CorticalFlow).**\
> **A1-1**: While ODE based formation has been widely used for diffeomorphic (image/surface) deformation, our method is different from CortexODE and CorticalFlow in following aspects:
>
> 1) Only CortexODE is based on “neural ODE” [1]. It uses the **_entire_** neural network (NN) to represent the continuous dynamics of the vertex trajectory. Deformation predictions are computed by a black-box differential equation solver, involving $T$ forward propagations (i.e., time complexity $O(T)$, T: # of steps). In contrast, our approach bypasses the need for $T$ forward propagations of the network by directly sampling VFs.
>
> 2) CorticalFlow uses _several_ NNs to predict a chain of 3 deformations to reconstruct a surface in a coarse-to-fine manner. It requires stage-wise training for each NN and may incur errors in coarse mesh template. In contrast, our method trains _one_ deformation NN on a fine initialization surface.
>
> 3) CortexODE & CorticalFlow do not consider physical relations between multiple cortical surfaces and need to train different models to reconstruct the WM and pial surfaces separately. In contrast, our method jointly reconstructs the WM, pial, and midthickness surfaces with cortical thickness estimation in an end-to-end learning framework.
>
> [1] Chen RT et. al. Neural ordinary differential equations (NeurIPS 2018)
>
> **C1-2: The losses are commonly used in this task**\
> **A1-2**: SOTA CSR methods and their losses have been discussed in Table 1 & Sect. 2. They only used Chamfer, MSE, or edge loss. Our method incorporates multiple new regularization losses, including $L_{dist}$, $L_{cyc}$, and $L_{ss}$, which improved the surface reconstruction quality as demonstrated by the ablation studies (See **A3-2** below for explanation of each loss).
>
> **C1-3: Technical details in the paper seem not to have a wide applicability in other areas**\
> **A1-3**: 1) The benchmark shows the effectiveness (ASSD \~0.1mm, SIF \~0.02%) and efficiency (\~2s) of our method. It offers a fast alternative for large-scale neuroimage studies (e.g., neurodegenerative diseases). 2) It can be applied to reconstruct other structures from CT/MRI (Lines 76-77, [26,27,40]) and help downstream tasks, such as heart mesh reconstruction for fluid dynamics simulation [2].
>
> [2] Attar R, et. al. 3D cardiac shape prediction with deep neural networks: simultaneous use of images and patient metadata. (MICCAI 2019)
>
> **C1-4: The paper is like an improvement work of CortexODE (initialize midthickness surface)**\
> **A1-4**: Our method is built upon many pioneering works in image registration and cortical surface reconstruction. In addition to the initialization of a midthickness surface, the joint reconstruction of both WM and GM surfaces under multiple regularization terms improved the robustness, accuracy, and computational efficiency of cortical surface reconstruction, as demonstrated by the extensive experimental results.
>
> Please refer to my response "_A1-1 to Reviewer 7qip_" for detailed comparison on surface initialization, and "_A1-1 above_" for ODE solution.
>
> **C2: Other papers’ visualization does not have equally distributed errors. How is Fig. 4 constructed to demonstrate results?**\
> **A2**: 1) The Nilearn package was used to map geometric errors onto surfaces. Others might use different tools with better shading and lighting. 2) We used more scales of color bar and thus more details were rendered. 3) When zooming in, we can observe that errors were not evenly distributed.
>
> In all, the lighter color in the error map signifies our method's lower geometric error. We will improve the visualization to highlight differences of the results obtained by different methods.
>
> **C3: Loss function\
> C3-1: S4, why is SIF of L-Pial bad but SIF of L-WM good?**\
> **A3-1**: We postulated that this was caused by the fact that pial surfaces are more convoluted than WM surfaces and contrast on the GM/CSF interface is worse than that of WM/GM interface. When the model was trained with the Chamfer loss alone, the VF might not be smooth enough, incurring self-intersection.
>
> **C3-2: Table 3b, using auxiliary losses gets worse performance. Why use them?**\
> **A3-2**: _Topology correctness and accuracy are equally important_ in the cortical surface reconstruction. To obtain topologically correct surfaces, we may need to sacrifice the accuracy.
> + Chamfer loss $L_{ch}$ only considers the coordinates of vertices with no topology constrains, resulting in topological errors in highly curved regions (setting S4).
> + Trajectory loss $L_{dist}$ helps enforce the midthickness surface to reside between WM and pial surfaces, avoid surface intersection, and mitigate challenges in learning large deformations (Lines 144-149). SIF was significantly reduced, particularly on the complex pial surface (setting S3).
> + Cycle loss $L_{cyc}$ and symmetric similarity loss $L_{ss}$ promote invertibility of the VFs to couple multiple CSR and enable cortical thickness estimation from the trajectory. S2 & S1 settings showed that surface quality further improved. Geometric accuracy changes were possibly attributed to the smoothing effect of these loss terms.
> + Normal consistency loss $L_{nc}$ discourages neighboring faces from forming sharp angles as vertex is deformed independently through sampled velocity from VFs. S0 setting showed that these regularization terms improved both geometric accuracy and surface quality.
>
> **C4: Any potential negative societal impact of the work?**\
> **A4**: We use 2 renowned public datasets collected with standards that control sex, age, and health status etc. But there may be oversights in terms of ethnicity and socioeconomic status, potentially putting underrepresented groups at a disadvantage in the study. We will discuss the potential impact and the need for more diverse datasets for validation.

---

> > ### Comment · Reviewer_DGUA · 2023-08-16
> >
> > Thanks for the rebuttal from the reviewers.
> > - In terms of running time of CortexODE, O(T) is not slow, since T is the hyperparameter and does not scale with respect to data. Thus, I think the difference is minor and I still think the work is incremental of CortexODE with some modifications.
> > - In terms of the losses functions, as the regularizations does not really improve the performance of the model (as stated in Table3), I think this proposals are inessential as well.
> > - In terms of the applicability of the method, since the method is specifically designed to address the cortical surface reconstruction with midthickness, I doubt that any other domains also have midthickness as initialization similar to cortical domains (including works that the authors have cited).
> >
> > In summary, I am afraid that the proposed method is rather the improvement of previous cortical surface reconstruction methods (CortexODE, CorticalFlow) and solve only the cortical surface problem, though have some minor modifications. Therefore, the applicability of the paper is narrow, and the paper can be more suitable to other domain specific conferences, so I decide to remain my score.

---

> > > ### Author Response · Authors · 2023-08-16
> > > **Response**
> > >
> > > Thanks for the comments.
> > >
> > > 1\. Running time of and comparison with CortexODE.
> > >
> > > We analyzed the time complexity of different algorithms for highlighting their differences. We agree that O(T) does not scale w.r.t data. However, the choice of T may significantly impact the surface _topology quality and geometric accuracy_, as stated in CortexODE paper.
> > >
> > > Our network is _not_ based on NODE. \
> > > Our method _jointly_ reconstructs pial, WM, and midthickness surfaces in a united network, while CortexODE trains two networks separately for reconstructing WM and pial surfaces. \
> > > Our method has achieved significant _improvement_.
> > >
> > > 2\. Loss function usage.
> > >
> > > The regularization terms _did_ improve performance, especially in terms of _topology quality_ (Table 3).
> > >
> > > We would like to highlight that **_the cortical surface topology is of great importance_**. Most of the cortical surface reconstruction studies, if not all, evaluate the reconstructed surfaces in terms of both **_topological correctness_** and **_accuracy_**.
> > >
> > > The loss functions adopted in our study facilitate accurate reconstruction of the cortical surfaces with minimal topological errors, and _our model built with all of them has achieved better performance than alternatives by a large margin in nearly all metrics_ (Table 2 main experimental results).
> > >
> > > 3\. Applicability.
> > >
> > > * Cortical surface reconstruction and thickness measurement are vital in _brain morphometry analysis_ and have broad applications in studies of brain development [1], brain aging [2], brain diseases [3], and brain changes [4]. Our method offers a _fast_ means with _improved geometric accuracy and topology quality_ for cortical surface analyses in _large-scale neuroimage studies_.
> > >
> > > * Our method can also be applied to _studies of other organs_, such as the heart. Particularly, our method can be used to reconstruct the inner and outer surfaces of the left ventricle and estimate its thickness.
> > >
> > > In summary, our method improves cortical surface reconstruction and has wide applications in imaging studies of the brain and heart. Moreover, our study fits the “_machine learning for sciences (e.g., health)_” and “_neuroscience and cognitive science_” categories of NeurIPS’ call for papers.
> > >
> > > [1] Gilmore JH, Knickmeyer RC, Gao W. Imaging structural and functional brain development in early childhood. Nature Reviews Neuroscience. 2018 Mar;19(3):123-37.
> > >
> > > [2] Lee J, Burkett BJ, Min HK, Senjem ML, Lundt ES, Botha H, Graff-Radford J, Barnard LR, Gunter JL, Schwarz CG, Kantarci K. Deep learning-based brain age prediction in normal aging and dementia. Nature Aging. 2022 May;2(5):412-24.
> > >
> > > [3] McColgan P, Joubert J, Tabrizi SJ, Rees G. The human motor cortex microcircuit: insights for neurodegenerative disease. Nature Reviews Neuroscience. 2020 Aug;21(8):401-15.
> > >
> > > [4] Bethlehem RA, Seidlitz J, White SR, Vogel JW, Anderson KM, Adamson C, Adler S, Alexopoulos GS, Anagnostou E, Areces-Gonzalez A, Astle DE. Brain charts for the human lifespan. Nature. 2022 Apr 21;604(7906):525-33.

---

> > ### Comment · Reviewer_DGUA · 2023-08-18
> >
> > Thank you for your rebuttal.
> >
> > - Since you also agree with me that O(T) does not scale wrt data, you can tune it to have a comparable time with your method. Moreover, CortexODE took only around 3-4s to infer on cases (fig 7 in CortexODE), your proposed approach take 4.5s to infer.
> >
> > - Furthermore, while your surface reconstruction time takes 1.5s to reconstruct, theirs only take less than 1s. I understand that the variance in running time could be due to hardware difference. However, based on those observations, your proposed approach does not run faster than CortexODE in general.
> >
> > - For the loss functions, based on your report results from Table 3, except the SIF of WM by using only Chamfer with terrible score (2.522), the other added losses only slightly improve the performance, e.g. ~0.01 in geometry metric and ~0.001 in SIF metrics.
> >
> > - Finally, about the applicability, I want to clarify that your work is more suitable in domain specific conferences, such as brain/medical-related conferences rather than NeurIPS. The reason is that your proposed method is specifically designed to solved the cortical surface reconstruction with midthickness (even this idea is not novel as well). I agree with the author that it can be applied to similar medical problem, such as heart problem as you pointed out, but the method has not been tested in that problem. Even some of your works you have cited in the discussion are in domain-specific venues.
> >
> > - In summary, while I appreciate the improvement in performance (probably come from the midthickness initialization) and the proposed paper solve the cortical surface reconstruction, I think that the technical contribution is limited and the paper solve a too narrow problem, which is hard to transfer to other problem. Therefore, I maintain my score after the discussion.

---

> > > ### Author Response · Authors · 2023-08-19
> > > **Response to Reviewer DGUA**
> > >
> > > Thanks for the comment.
> > >
> > > 1. Time
> > > + The choice of T may significantly impact the surface topology quality and geometric accuracy, as stated in CortexODE paper (_Fig. 8_). _A smaller T may lead to worse results_. CortexODE proposes to use a larger T in inference than in training.
> > > + CortexODE reconstructs WM and pial surfaces **sequentially**, thus their inference time (for the network) the reviewer referred to should be **doubled** for a fair comparison. In contrast, we reconstruct three surfaces (WM, midthickness, pial surfaces) in one forward propagation, thus our method’s computational time is shorter. FYI, we benchmarked the time on the same machine (Lines 306-310).
> > >
> > > 2. Pial and WM surfaces are **equally important**. The improvement should be interpreted **jointly** for _both_ surfaces. Our method achieves a balance between accuracy and topology quality and significantly outperforms alternatives.
> > >
> > > 3. Please refer to our previous explanation on applicability. We respect the reviewer’s personal opinions.

---

> > > > ### Comment · Reviewer_DGUA · 2023-08-20
> > > >
> > > > Thanks for your response.
> > > >
> > > > **Time consumption:**
> > > >
> > > > (1) The WM refinement of CortexODE can be very fast, since their WM initialization is very close to the pseudo ground truth, which means they don't necessarily require large step size to converge.
> > > >
> > > > (2) They applied Euler method to integrate velocity field, which can be super fast during inference. In practice, they only used step size = 0.1 (**10 steps**) during inference.
> > > >
> > > > (3) What's more, the **majority** time consumption of the whole reconstruction will be topology correction instead of surface reconstructions. And the topology correction of mid-thickness should be more complicated than white matter because more holes and bridges are wrongly reconstructed via segmentation.
> > > >
> > > > **Therefore, the time consumption gap between your proposal and CortexODE is not as significant as it appears to be.**
> > > >
> > > > **Midthickness initialization:**
> > > >
> > > > (1) Yes, surface initialization is very important. The idea of template-based mesh reconstruction is not novel. The idea of taking segmentation results as initializations is not novel. Conceptually, a good surface initialization leading to better reconstruction is not novel.
> > > >
> > > > (2) Even, taking midthickness as initialization is not that new in cortical surface reconstruction problem. In the implementation of CortexODE, they already used the *inflated white matter* (https://github.com/m-qiang/CortexODE/blob/ca7a59de261c3ca8cc69fb0c3bc845f0f862b28b/eval.py#L179-L183) as the initialization of grey matter reconstruction, which is very similar to midthickness initialization in your proposal.
> > > >
> > > > (2) Direct estimation of cortical thickness together with mesh-based cortical surface reconstruction problem is new. But I suppose the effect of L_dist is **largely overrated** in the surface reconstruction task. Compared with solely using CH, using L_dist can only benefit SIF. But there are other effective efforts can be done to improve it. I am curious, what if you don't use L_dist, but apply all other loss terms?

---

> > > > > ### Author Response · Authors · 2023-08-21
> > > > > **Response to Reviewer DGUA**
> > > > >
> > > > > Thanks for the comments. We also appreciate the reviewer for reviewing our paper and dedicating time to the discussions.
> > > > >
> > > > > **1. Time consumption**
> > > > >
> > > > > **First**, we want to present a few **facts** in response to the reviewer's comments:
> > > > > + In CortexODE paper (Page 6, Sect. IV, A-2), they say “the model is discretized by an Euler solver with total numerical steps **N = 10** and step size h = 0.1 for training. For inference, we set **N = 20** for all datasets to reduce self-intersections.”
> > > > > + In CortexODE paper (Page 9, Sect. IV, C-2), the authors analyzed the impact of T and ODE solvers on topology and accuracy (Fig. 8). _A smaller T may lead to worse results_.
> > > > > + According to CortexODE paper (Page 5, Sect. III, C) and their cited [25] (P.-L. Bazin and D. L. Pham, “Topology correction of segmented medical images using a fast marching algorithm,” CMPB’2007), _the time is proportional to the dimensions of data volume_ rather than the implicit surface.
> > > > >
> > > > > **Second**, we benchmarked all baseline methods on the same machine, which supported the above statements. We have analyzed the _main_ computational time in Lines 306-310 in our main paper. _If needed, we can add more details in supplementary materials_.
> > > > >
> > > > > **Third**, we did _not_ mix the topology correction and network runtime in our paper.
> > > > >
> > > > > **Fourth**, in our paper Lines 147-149, we explained “_the distance between the midthickness surface and the pial surface can help prevent topology errors in cases of deep sulci, as this layer still provides clear separation between them_”. _If needed, we can show visual results_. We did check the topology correctness of the reconstructed midthickness surface.
> > > > >
> > > > > In summary, **we appreciate the reviewer for acknowledging that the time cost of our method is lower**. We respect the reviewer’s **personal opinions** and have no comments on other **speculations**.
> > > > >
> > > > >
> > > > > **2. Midthickness initialization**
> > > > >
> > > > > **First**, our experiment results have demonstrated the effectiveness of our initialization method in terms of both surface location and quality, **which has not been demonstrated by prior methods**.
> > > > >
> > > > > **Second**, please refer to our response "A1-1 to Reviewer 7qip" for detailed comparison/explanation on surface initialization w.r.t. CortexODE. In short:
> > > > >
> > > > > + CortexODE needs **separate** initialization surfaces for WM and pial surfaces. In contrast, we **coupled** multiple surface reconstructions in a united framework, requiring **one** midthickness surface initialization.
> > > > >
> > > > > + CortexODE _deforms the WM surface outward along the normal direction $\vec{n}$ by a manually selected distance $\rho$_. It is **not necessarily optimal for all cortical regions** b/c the thickness distribution is not uniform across the entire surface, and **$\rho$ has to be chosen empirically** (i.e., not necessarily midthickness surface). In contrast, in our method, each vertex is determined by local WM and GM structures, resulting in a **more precise approximation** of the true midthickness surface, and **no hyperparameter $\rho$ is needed**.
> > > > >
> > > > > **Third**, _we thank the reviewer for acknowledging that integrating direct thickness estimation into cortical surface reconstruction is **new**_.
> > > > >
> > > > > **Fourth**, we claimed that “_trajectory loss_ $L_{dist}$ _helps enforce the midthickness surface to reside between WM and pial surfaces, avoid surface intersection, and mitigate challenges in learning large deformations_”. We do _not_ understand which part is “_largely overrated_” as the reviewer also agrees that it “benefits SIF”.
> > > > >
> > > > > **Last**, in our preliminary experiments, we found that trajectory loss was _very/most efficient_ to reduce SIF and helped convergence, **thus** _we conducted ablation studies of the loss terms in the current order_. If needed, _we can add more details_ in the supplementary materials.

---

### Official Review · Reviewer_YgxF · 2023-07-07

**Soundness:** 3 good
**Presentation:** 3 good
**Contribution:** 2 fair
**Rating:** 6
**Confidence:** 3

**Summary:**

The paper focuses on the cortical surface-mesh reconstruction in Magnetic Resonance Imaging (MRI) data. The goal is to reconstruct both the inner and outer surfaces of the cortical region. The proposed method begins by estimating an intermediate midthickness layer, which is positioned halfway between the inner and outer surfaces. This midthickness layer serves as the initialized surface mesh for the reconstruction process.

To evaluate the effectiveness of the proposed method, the authors present results obtained from two neuroimage MRI datasets. These datasets likely consist of MRI scans of the brain, specifically targeting the cortical region. The authors use these datasets to showcase the performance of their surface reconstruction technique.

By employing their proposed method, the authors aim to demonstrate the ability to accurately reconstruct the cortical surfaces from MRI data. This can be valuable for various applications in neuroscience and neuroimaging research, such as brain mapping, cortical thickness analysis, and studying cortical abnormalities or diseases.

**Strengths:**

The paper presents a significant advancement in cortical mesh reconstruction and thickness estimation. The key innovation lies in the incorporation of a constraint that ensures the model predicts both the inner and outer meshes while considering a plausible distance representing the cortical thickness between the white matter and gray matter. While previous studies have explored the use of ODE for cortical mesh reconstruction, the proposed method employs a distinct approach by leveraging this technique to obtain two mesh surfaces and incorporates several loss functions to optimize the results.

Overall, the paper introduces a valuable contribution to the field by addressing the challenge of accurately reconstructing cortical meshes and estimating cortical thickness. The novel constraint and the unique application of ODE set this approach apart from previous works.

**Weaknesses:**

The choice of spatial resolution for applying Marching Cubes on the Signed Distance Function (SDF) volume has a direct impact on the granularity of the resulting mesh, specifically in terms of the number of vertices generated. Increasing the number of vertices can potentially reduce the geometry reconstruction error, but this comes at the expense of higher computational requirements and increased memory consumption. Surprisingly, the paper lacks any explanation or analysis regarding this crucial aspect.

It would be beneficial for the paper to provide a detailed discussion or analysis concerning the trade-off between spatial resolution, mesh granularity, and the associated computational and memory costs. This would help readers understand the implications of different spatial resolutions on the quality of the reconstructed geometry. Additionally, examining the potential impact on downstream applications or comparing the performance across different resolutions could provide valuable insights for future research and practical implementations.

**Questions:**

- In your framework, after computing the average of WM and GM SDFs, the Marching Cubes algorithm is used for mesh reconstruction. What are the advantages of incorporating ODE?

- What is the average number of vertices in the initialized mesh, and how does it affect the output metrics?

- The caption of Figure 2 (a) does not adequately explain the graphs in the Figure. Please provide a clearer description in the text.



**Limitations:**

The paper should thoroughly address the limitations associated with converting the segmentation map into an SDF volume, emphasizing how the resolution of the 3D segmentation volume affects accuracy. Additionally, discussing the potential implications or consequences of these limitations for downstream applications is crucial.

To provide a comprehensive analysis, it is essential to explicitly mention the failure cases and limitations of the proposed method. Discussing scenarios in which the method might not perform optimally or cases where it is not suitable would enhance the paper's credibility and assist readers in understanding the method's boundaries and potential weaknesses.

---

> ### Author Rebuttal · Authors · 2023-08-08
>
> Thanks for your comments and suggestions. We will revise our paper accordingly.
>
> **C1: Surface granularity\
> C1-1: The choice of spatial resolution for applying Marching Cubes on the Signed Distance Function (SDF) volume directly impacts resulting mesh granularity. More vertices can reduce the reconstruction error but increase computation cost. A detailed discussion/analysis on the trade-off between spatial resolution, mesh granularity, and the associated computational costs would clarify implications of different spatial resolutions on the geometry quality.**
>
> **A1-1**: The standard brain MRI image resolution is $256^3$. As a proof of concept, we apply Marching Cubes (MC) on $256^3$ and up-sampled $512^3$ WM SDF. The reconstructed WM surface ASSD is reduced from 0.265mm to 0.254mm, but the computation time increases from 0.23s to 2.45s for MC and from 2.8s to 19.3s for topology correction (i.e., \~7X totally). The similar trend is observed on the pial surface with lower geometric accuracy. We also increase the mesh granularity by mesh subdivision (\~1s for a mesh with 130K vertices), but the ASSD barely changes (0.256mm). However, reducing the resolution by down-sampling SDF substantially increases the reconstruction error (especially for the pial surface). In summary, simply applying Marching Cubes to SDF at higher resolutions does bring small improvement on geometric accuracy but incurs heavier computational cost; it may degrade significantly or fail if the surface initialization is obtained based on SDFs at lower resolutions.
>
> **C1-2: What is the avg. number of vertices in the initialized mesh? How does it affect the output metrics? Examining the potential impact on downstream applications could provide valuable insights for future research and practical implementations.**
>
> **A1-2**: Following previous works, we used meshes with \~130K vertices in our experiments.
>
> To test the impact of the number of vertices on the downstream tasks, we simplify the meshes by merging neighboring faces (\~60K) and subdivide the meshes by dividing each face into four faces (\~520K) and utilize them to train different networks. Due to time limit, we randomly choose a subset from ADNI-1 dataset (300, 20, 80 for training, validation, and test, respectively) to conduct the experiments. The ASSD results are as follows:
> ||60K|130K|520K|
> |-|-|-|-|
> |L-WM|0.122|0.074|0.067|
> |L-pial|0.201| 0.156|0.154|
>
> **C2: In your framework, after computing the average of WM & GM SDFs, the Marching Cubes algorithm is used for mesh reconstruction. What are the advantages of incorporating ODE?**
>
> **A2**:
> **_First_**, higher geometric accuracy. Marching Cubes (MC) works on voxel-level but cortical surface reconstruction (CSR) requires sub-voxel accuracy. Simply applying MC yields worse ASSD (mm) (WM: \~0.265, pial: \~0.323) than our method (WM: \~0.071, pial: \~0.136).
> **_Second_**, better topological quality. Due to the partial volume effect and cerebral cortex's thin and highly-folded pattern (Lines 138-139), the contrast on the gray matter (GM)/cerebrospinal fluid (CSF) interface is very low. Directly applying MC to reconstruct pial surface inevitably generates topology errors (SIF: \~8%). In contrast, the ODE formulation provides stronger theoretical guarantees for diffeomorphic deformation and we deform an initialization surface with good topology to the pial surface, resulting in better topological quality (SIF: \~0.02%).
> **_Third_**, by coupling multiple CSR within a united framework, the geodesic trajectory of each vertex serves as an estimation of the cortical thickness.
>
> **C3: Fig. 2a caption does not adequately explain the graphs in the Figure. Please provide a clearer description in the text.**
>
> **A3**: We will add more description. This sub-figure illustrates the surface reconstruction accuracy (Y-axis) with initialization surfaces at different locations (X-axis) for WM and pial surfaces, respectively (using PialNN method). The X-axis represents the initial surface generated from WM SDF ($K_W$), GM SDF ($K_G$), or midthickness SDF ($K_M$). The Y-axis represents the ASSD metric of the reconstructed surface.
>
>
> **C4: Authors should mention the failure cases and limitations of the proposed method, e.g., scenarios where the method might not perform optimally or is not suitable. It would enhance the paper's credibility and help readers to understand the method's boundaries and potential weaknesses.**
>
> **A4**: The standard deviation is very small in our experimental results. We carefully checked the test results and did not find "visually" failed cases. Nevertheless, we will visualize results with the worst performance in terms of accuracy and topology correctness in the supplementary file. We postulated that the good generalization performance should be attributed to the fact that our model was trained on large datasets and both the training and testing datasets consisted of high-quality T1 MRI scans of adult subjects.
>
> There are a few limitations of our method:
> + Despite achieving a low self-intersection ratio and improved cortical surface reconstruction accuracy, our method can be further improved by adopting and exploring post-processing methods and designing new loss functions to further minimize SIF and improve surface quality.
> + While focusing on cortical thickness estimation in this paper, we recognize the value of incorporating other cortical attributes like surface area and sulci depth into CSR and analysis tasks. These attributes could serve as supplementary constraints, enhancing overall reconstruction quality.
> + We demonstrate the correlation between the estimated cortical thickness and that of FreeSurfer in this paper. We should conduct more analyses on a broader range of subjects (e.g., health subjects and subject with dementia or Alzheimer’s disease) to evaluate the effectiveness of the proposed method and its limitations.
>
> We will add a section to discuss limitations and broader/societal impact.

---

> > ### Comment · Reviewer_9EVU · 2023-08-19
> >
> > Thanks for the authors' detailed response, my concern has been resolved, and I will keep my original weak accept score.

---

### Official Review · Reviewer_9EVU · 2023-07-17

**Soundness:** 3 good
**Presentation:** 3 good
**Contribution:** 3 good
**Rating:** 6
**Confidence:** 4

**Summary:**

This paper presents a deep-learning-based approach for cortical surface reconstruction (CSR) with topology constraints to ensure the physical meaning of reconstructed surfaces. Experiments on public benchmarks show that the proposed approach outperforms current state of the art networks.

**Strengths:**

The overall presentation is clear and easy to follow.

In cortical surface reconstruction, the reservation of the reconstructed surface geometry is a crucial and challenging part. This paper aims at tackling the problem by generating a mid-thickness surface with topology constraints, which is intuitive and well-motivated.

The experimental results seem to be encouraging and the proposed method outperforms SOTAs by a clear margin.

**Weaknesses:**

The claimed contributions are over-stated, and several related works seem to be ignored and would potentially lead to misunderstanding for the readers.
"the cortical thickness estimation is neglected in all existing DL-based CSR methods" is simply not true. There are plenty of related methods on cortical surface reconstruction/prediction/parcellation that consider cortical attributes (cortical thickness, surface area, cortical folding, diffusivity, structural and functional connectivity, etc) in their frameworks [1,2]. In [3], cortical thickness was already considered as a constraint during the network's supervised learning process, and their network was also designed to predict *both* the surfaces (inner and outer) and cortical thickness at the same time.
Since all these works are closely related, the authors should at least be aware of and mention or discuss the relations and differences between these methods and their proposed one.

[1] Wu et al. Surface-based analysis of the developing cerebral cortex. Advances in Magnetic Resonance Technology and Applications, 2021.
[2] Zhao et al. Spherical Deformable U-Net: Application to Cortical Surface Parcellation and Development Prediction. IEEE Transactions on Medical Imaging, 2021.
[3] Liu et al. Deep modeling of growth trajectories for longitudinal prediction of missing infant cortical surfaces. International Conference on Information Processing in Medical Imaging (IPMI), 2019.

**Questions:**

N/A

**Limitations:**

The authors did not discuss their method's limitations, or the potential future directions.

---

> ### Author Rebuttal · Authors · 2023-08-09
>
> **C1: Contributions over-stated, several related works ignored, and potential for reader misunderstanding.\
> The statement "the cortical thickness estimation is neglected in all existing DL-based CSR methods" is incorrect. Related methods on cortical surface reconstruction/prediction/parcellation consider cortical attributes (thickness, area, structural and functional connectivity, etc.) in [1,2].\
> In [3], cortical thickness was considered as a constraint in network learning, and it also predicts both inner and outer surfaces and thickness concurrently. The authors should acknowledge and discuss the relations and differences between these methods and their proposed one.**
> > [1] Wu et al. Surface-based analysis of the developing cerebral cortex. (AMRTA 2021)\
> > [2] Zhao et al. Spherical Deformable U-Net: Application to Cortical Surface Parcellation and Development Prediction. (TMI 2021)\
> > [3] Liu et al. Deep modeling of growth trajectories for longitudinal prediction of missing infant cortical surfaces. (IPMI 2019)
>
> **A1**: Thanks for your suggestions. We will revise the paper as suggested.
>
> First, we would like to clarify that the statement ("the cortical thickness estimation is neglected in all existing **_DL-based CSR_** methods") intended to depict the methods dedicated to “DL-based cortical surface reconstruction (CSR)” rather than the “entire pipeline of the cortical surface analysis”. We do not focus on non-DL methods or toolboxes/pipelines that enable various tasks in brain MRI analysis in separate steps (i.e., independent solutions for topology correction, cortical thickness/area estimation, and parcellation, etc.). This paper focuses on CSR with one important attribute, cortical thickness. We will further clarify our scope in the main paper.
>
> Second, we want to elaborate the relations and differences between the mentioned methods and ours.
>
> 1. [1] outlines the entire pipeline for analyzing developing cerebral cortex, encompassing the following four steps:
> + (1-a) Cortical topology correction. A learning-based method is presented to adaptively correct detected topological errors, whereas we use a topology correction algorithm on a signed distance function for initializing the midthickness surface.
> + (1-b) Cortical surface reconstruction. A two-step non-DL approach is presented to generate the inner and outer cortical surfaces _separately_. Specifically, a tessellation method is used to extract inner surface from the WM segmentation map, and then the inner surface is deformed to the outer surface driven by an external force derived from the Laplace's equation and an internal force to keep the surface tight and smooth, along with some constrains on the spherical topology. The middle surface is then reconstructed as the average of inner and outer surfaces. On the contrary, our DL-based method _jointly_ reconstructs multiple cortical surfaces in a united framework.
> + (1-c) Cortical surface-based measurement on the reconstructed surfaces. Our main focus is on the cortical surface reconstruction from imaging data.
> + (1-d) Cortical surface registration, parcellation, modeling, and atlas construction. The rest tasks are out of scope of this work and could be a future direction.
>
> 2. [2] introduces a Spherical Deformable U-Net that operates on surface meshes, focusing on two tasks: cortical surface parcellation (a vertex-wise classification/segmentation task) and cortical attribute map development prediction (a vertex-wise dense regression task). In contrast, our method primarily focuses on cortical surface reconstruction and thickness estimation.
>
> 3. [3] proposes a method for longitudinal prediction of cortical surfaces using a spatial graph convolutional neural network. Note that their method is not designed for CSR from brain MRIs but instead predicting longitudinal surface changes within the same individual. The initial cortical surfaces are generated by a toolbox as input to their framework. While they do incorporate a thickness constraint using the Euclidean distance between paired vertices on inner and outer surfaces, the predicted thickness is compared with the ground truth thickness by L2 norm. On the contrary, our approach traces the geodesic distance in the vertex deformation trajectory as thickness estimation, without utilizing ground truth thickness as supervision.
>
> In summary, we will discuss our method’s relatedness to existing works in cortical surface analysis and clarify the focus and scope of our work.
>
> **C2: The authors did not discuss their method's limitations, or the potential future directions.**
>
> **A2**: Thanks. We will add a paragraph discussing our method’s limitations and future directions. For example,
>
> + Despite achieving a low self-intersection ratio and improved cortical surface reconstruction accuracy, our method can be further improved by adopting and exploring post-processing methods and designing new loss functions to further minimize SIF and improve surface quality.
>
> + While focusing on cortical thickness estimation in this paper, we recognize the value of incorporating other cortical attributes like surface area and sulci depth into CSR and analysis tasks. These attributes could serve as supplementary constraints, enhancing overall reconstruction quality.
>
> + We demonstrate the correlation between the estimated cortical thickness and that of FreeSurfer in this paper. We should conduct more analyses on a broader range of subjects (e.g., health subjects and subject with dementia or Alzheimer’s disease) to evaluate the effectiveness of the proposed method and its limitations.

---

### Official Review · Reviewer_7qip · 2023-07-17

**Soundness:** 3 good
**Presentation:** 3 good
**Contribution:** 2 fair
**Rating:** 4
**Confidence:** 4

**Summary:**

This paper proposed a method for cortical surface reconstruction by deforming a middle layer inwards and outwards to be the gray matter and white matter surface respectively. First, segmentation network and topology correction are used to extract the initial surface. Then, three velocity fields are predicted to deform the initial surface to be the final white matter and gray matter. The experiments demonstrate the proposed method and the design.

**Strengths:**

1. The idea of deforming the middle surface to generate white matter and gray matter is interesting.
2. The experiments are sound and demonstrating, including the comparison between the SOTA methods and the ablation study for the components.



**Weaknesses:**

This paper seems to overstate its contribution. First, it claims that it devises an efficient and reliable approach for initialization. However, the initialization step is basically following CortexODE, where the CortexODE predicts the white segmentation, and initializes the WM surface, this paper initializes the midthickness surface. The pipeline of initialization is the same as proposed by CortexODE. Second, the diffeomorphic deformation. This paper claims that they design a new numerical computation strategy to calculate the diffeomorphic deformation trajectory, where the deformation is the accumulated velocity along the several time steps. The CortexODE also utilizes explicit fixed-step methods for ODE discretization instead of the scaling and squaring methods.

The idea of utilizing the middle state to generate the final result is also not new, a similar approach can be found in [1].

[1] Mok, Tony CW, and Albert Chung. "Fast symmetric diffeomorphic image registration with convolutional neural networks

**Questions:**

Figure 1 is confusing to me, For the cyclic loss on the GM as the example, why the forward and backward deformation is from the same velocity field? In Eq. 2, they should use different velocities.

**Limitations:**

Please try to solve the questions and weakness above.

---

> ### Author Rebuttal · Authors · 2023-08-08
>
> Thanks for your comments and suggestions.
>
> **C1: This paper seems to overstate its contribution.\
> C1-1: First, the initialization pipeline is following/the same as CortexODE (seg + WM surf.).**
>
> **A1-1**: Our objectives and emphases are different from them.
> + First, our method jointly reconstructs **_both_** the inner and outer cortical surfaces based on an initialization of the midthickness surface, different from CortexODE that relies on the WM segmentation for **_soely_** WM surface initialization and reconstruction. Our experiment results have demonstrated the effectiveness of our initialization method in terms of both surface location and quality (Lines 139-142; Fig. 2(a); Table 3, M1, M2), which has not been demonstrated by prior methods.
>
> + Second, CortexODE needs to train a secondary model for pial surface reconstruction, entailing an additional step for pial surface initialization. Specifically, they deform the WM surface outward along the normal direction $\vec{n}$ by a manually selected distance $\rho$. However, this strategy is _not necessarily optimal_ for all cortical regions because the thickness distribution is not uniform across the entire surface. In contrast, our midthickness surface initialization leverages both WM and GM segmentation maps (Lines 167-169). Each vertex is determined by local WM and GM structures, resulting in a more precise approximation of the true midthickness surface.
>
> **C1-2: Second, the diffeomorphic deformation is the accumulated velocity along several time steps. CortexODE also utilizes explicit fixed-step methods for ODE discretization instead of the scaling and squaring methods.**
>
> **A1-2**: To ensure diffeomorphic deformation (i.e., stronger theoretical guarantees), a natural framework is to formulate it as a flow ODE, modeled as an initial value problem depicted in Eq. (1), and parameterize a set of learnable deformations. Although the problem formulation and high-level idea of numerical solutions (e.g., Euler method) are similar, our concrete solution distinctly differs from that of CortexODE.
>
> CortexODE employs the **_entire_** neural network to represent the continuous dynamics of the vertex trajectory, which is why it is called “neural ODE” [1] based method. The output of the network is computed using a black-box differential equation solver and provides deformation predictions at each time step. The memory cost is $O(1)$ but the time complexity is $O(T)$ (involving $T$ forward propagations), where $T$ is the number of steps.
>
> In contrast, our approach utilizes a neural network to learn velocity fields (VFs) and bypasses the need for $T$ forward propagations of the network for velocity predictions by leveraging sampling the VFs $T$ times. Additionally, the integration operation is solely on sampled vertices (\~130K). This is different from typical scaling and squaring methods used in image registration that apply the integration to the whole volume grid (\~8M, i.e., 55 times larger), and therefore our method is more efficient.
>
> [1] Chen RT, Rubanova Y, Bettencourt J, Duvenaud DK. Neural ordinary differential equations. (NeurIPS 2018)
>
> **C2: Using the middle state to generate the final result is not new, a similar approach can be found in: [1] Mok, Tony CW, and Albert Chung. Fast symmetric diffeomorphic image registration with convolutional neural networks.**
>
> **A2**: We were aware of the referenced work, acknowledged it as [39], and discussed it in our main paper (Lines 119-128). Our method fundamentally distinguishes itself in three key aspects:
>
> + [1] is designed for 2D image registration, and the deformation field is computed on a pixel grid utilizing scaling and squaring method. A direct extension to 3D images would incur significant computational costs, and the deformation field computed on a voxel grid is not necessarily optimal for cortical surface reconstruction that deforms _vertices_ (i.e., sub-voxel level) instead of _voxels_.
> + [1] decomposes the transformation from image $Y$ to image $X$ into $\phi_{YX}^{(1)}(Y)=\phi_{YX}^{(0.5)}\circ\phi_{XY}^{(-0.5)}(Y)$ and estimates the “half” forward deformation $\phi_{YX}^{(0.5)}$ and the "half" inverse deformation $\phi_{XY}^{(-0.5)}$ by _integrating the velocity fields_ forward and backward respectively. That is, [1] deforms two known images to each other. In contrast, our method (i) computes _vertex-wise deformation_ from velocity fields directly; and (ii) deforms the midthickness surface to inner and outer surfaces and then deforms backwards to midthickness surface (Eq. 2, Fig. 2b).
> + Their symmetric deformation requires $\phi_{YX}^{(1)}(Y)\simeq X$ and $\phi_{XY}^{(1)}(X)\simeq Y$ respectively. On the contrary, our cycle loss ensures simultaneous alignment of $p_{\phi_W\circ\phi_G}\simeq p\simeq p_{\phi_G\circ\phi_W}$ for each vertex $p$ on the midthickness surface $S_M$.
>
> **C3: Figure 1 is confusing. Why is the forward and backward deformation from the same velocity field? In Eq. 2, they should use different velocities.**
>
> **A3**: Sorry for the confusion in Fig. 1. Your understanding regarding Eq. 2 was accurate. There was an error on the data flow arrows in Fig. 1. We realized this issue and rectified it in the supplementary materials (Fig.1). We will update Fig. 1 in the main paper.

---

> > ### Comment · Reviewer_7qip · 2023-08-15
> >
> > Thanks for the rebuttal from the reviewers. It solved some of my concerns, but still not enough for me to recommend the work.
> > First, the authors did a poor literature review, the work (Fast symmetric diffeomorphic image registration with convolutional neural networks) is originally designed for 3D medical images and achieves reasonable computational cost. However, the author points out that it was designed for 2D images, which is misleading and unprofessional.
> > Second, the answer in 1.1 didn't solve my concerns regarding the novelty of the initialization paradigm. The authors try to differentiate from the CortexODE in the following modules, however, my major concern is for the mid-surface initialization. In my view, Cortex ODE is the first to introduce the whole pipeline of initialization to the cortical surface reconstruction task, including the segmentation, SDF transformation, topology correction, and surface extraction. The initialization pipeline from this work follows exactly the same steps. I get that this paper wants to extract the mid-surface. However, the novelty in the initialization is weak.
> > Next, as mentioned in A1-2, Cortex ODE and this work, the diffeomorphic deformation are both accumulated along several time steps. The author should revise the third contribution, and highlight the different between the Corted ODE.
> > Last, even though, the detail of utilizing the middle state is different, the main idea behind is similar. But I'm glad the authors successfully utilize the method in this way.
> > In conclusion, I will remain my rating.

---

> > > ### Author Response · Authors · 2023-08-15
> > > **Clarification & Response**
> > >
> > > Thanks for the prompt response to our rebuttal.
> > >
> > > We apologize for the wrong statement that the method developed in “Fast symmetric diffeomorphic image registration with convolutional neural networks” was a 2D method.
> > >
> > > We would like to highlight that our method optimizes the diffeomorphic deformation field by minimizing the difference between the initialization midthickness surface/**_vertices_** and their target surfaces (WM and Pial surface). This is different from typical image registration methods that maximize image similarity gauged at a **_voxel grid_**.
> > >
> > > Sorry for the confusion. We did _not_ intend to claim the novelty of “WM segmentation + surface initialization + topological correction” in our method. Instead, we would like to highlight that the _usage of midthickness surface facilitates the coupled reconstruction of both the WM and Pial surfaces_ and that the _simultaneous optimization of the midthickness surface enables the estimation of the cortical thickness that improves the robustness of our method_ due to the regularization of “nonnegative thickness”.
> > >
> > > Third, our method models the diffeomorphic deformation with a dense deformation field that is optimized by minimizing the difference between vertices of the initialization and target surfaces. The dense deformation field enables to “_jointly_” optimize the deformation of all the surface vertices. This is different from CortexODE that adopts a solution of “Neural Ordinary Differential Equations” to optimize the deformation of individual vertices.
> > >
> > > We are confused by "_the middle state_” and appreciate any further clarification.

---

> > > > ### Comment · Reviewer_DGUA · 2023-08-16
> > > >
> > > > In terms of the third clarification:
> > > >
> > > > First, dense deformation field is applied by CorticalFlow and CorticalFlow++, could you please elaborate the technical difference between your proposed methods deformation field and theirs.
> > > >
> > > > Second, NODE doesn't optimize each individual vertices separately, the deformations of all vertices are governed one single ode function, represented by an MLP. In other words, the optimization variables are the parameters of MLP, instead of the deformations of separate vertices. Furthermore, training with the dense deformation field still all kinds of regularizations, which can also be applied in training NODE.

---

> > > > > ### Author Response · Authors · 2023-08-17
> > > > > **Response to Reviewer DGUA**
> > > > >
> > > > > Thanks for the comments.
> > > > >
> > > > > 1\. CorticalFlow and CorticalFlow++ use a _chain_ of deformable networks to predict a series of deformation fields that deform the initial mesh template to the target surface in a _coarse_ (surfaces with a smaller number of vertices) _-to-fine_ (surfaces with a larger number of vertices) manner. Since the deformation fields are _trained subsequently_, the final results are hinged on intermediate results whose errors may propagate in multiple steps.
> > > > >
> > > > > In contrast, our method trains _one_ neural network to deform a _well-initialized midthickness surface_ in an _end-to-end_ fashion.
> > > > >
> > > > > 2\. We agree that a single NODE model (i.e., a set of parameters) is optimized for all vertices but in an _implicit_ manner (b/c each vertex independently samples local image cubes as input).
> > > > >
> > > > > We agree that the regularization terms used for the dense deformation fields can be applied to NODE models. However, extra effort might be needed to design such regularization terms to be applicable to NODE models.

---

> > > > > > ### Comment · Reviewer_DGUA · 2023-08-18
> > > > > >
> > > > > > Thanks for you response.
> > > > > >
> > > > > > First, CorticalFlow and the second step of your proposal can be seen as templated based image registration methods that is only supervised on the cortical surface. Dense, Multi-level, and end-to-end-trained deformation field predictions are applied in the very beginning of DL-based image registration methods, such as VoxelMorph.
> > > > > >
> > > > > > Second, the error might to be propagate in multiple steps but also can be corrected.
> > > > > >
> > > > > > Third, good initialization is very important, and we always want a better initialization. However, personally speaking, a good initialization is somewhat trivial in the aspect of methodology, considering the very important step, i.e., topology correction, is not your contribution.

---

> > > > > > > ### Author Response · Authors · 2023-08-19
> > > > > > > **Response to Reviewer DGUA**
> > > > > > >
> > > > > > > Thanks for your comments.
> > > > > > >
> > > > > > > First, our method is built upon many pioneering works in image registration and cortical surface registration. _We mainly claimed the following contributions and conducted experiments to support them_.
> > > > > > >
> > > > > > > + The initialization of a midthickness surface facilitates the coupled cortical surface reconstruction.
> > > > > > > + Simultaneous optimization of WM, Pial, and middthicknesss surfaces enables the estimation of the cortical thickness that improves the robustness of our method.
> > > > > > > + Regularization terms ($L_{cyc}$, $L_{ss}$, $L_{dist}$, $L_{nc}$) improve performance (topology & accuracy).
> > > > > > > + One model of 3D CNNs with multi-channel input is designed to utilize information from multiple sources and can be trained end-to-end.
> > > > > > >
> > > > > > > Second, we agree that those errors might be corrected. But correction methods have to be developed.\
> > > > > > > Compared to CorticalFlow also using a spherical (i.e., genus-0) initialization surface, our method obtained results with significantly fewer topology errors (worst-case SIF: ours ~0.035% vs  CF ~0.149% on ADNI L-Pial).
> > > > > > >
> > > > > > > Third, thanks for agreeing with us that good initialization is very important. \
> > > > > > > "Topology correction" is a necessary step in most cortical surface reconstruction methods but not our focus in this paper. Our study has demonstrated the impact of the initialization meshes (WM/pial/midthickness) (Fig. 2(a) & Lines 138-151;  Table 3, M1, M2) and provided a means to achieve this goal (i.e., where & how).

---

### Official Review · Reviewer_2ocD · 2023-07-24

**Soundness:** 3 good
**Presentation:** 4 excellent
**Contribution:** 3 good
**Rating:** 7
**Confidence:** 4

**Summary:**

This paper develops a new deep-learning-based framework to jointly reconstruct the inner, outer, and their in-between (midthickness) surfaces and estimate cortical thickness. The proposed method has been evaluated on two neuroimaging datasets and shown superior cortical surface reconstruction performance. This work is potentially interesting to the computational neuroscience community and valuable for brain study.

**Strengths:**

The authors propose a novel deep-learning-based framework to robustly reconstruct the cortical surfaces with topological correctness. Different from existing methods, it explicitly couples the inner and outer surfaces by jointly learning three diffeomorphic flows. Instead of using mixed structures of CNN and GNNs/MLP, this work utilizes a single 3D CNN to leverage 3D brain MRI, a ribbon segmentation map, and a signed distance function altogether. Also, the diffeomorphic deformation trajectory is calculated by a new numerical computation strategy. Furthermore, the initialization of fine midthickness surfaces is effective, and thickness estimation is performed together with cortical reconstruction. Overall, this paper is well-written and easy to follow. The proposed method is novel and can potentially alleviate several limitations of existing methods. The experiments also demonstrate the effectiveness of the proposed method on two neuroimaging datasets.

**Weaknesses:**

The demographics for the two testing datasets are not illustrated, and experiments can be performed on more populations.
I am confused about the effectiveness of different losses in Table 3.


**Questions:**

See limitations.

**Limitations:**

Experiments can be performed on more populations (e.g., babies) to evaluate the effectiveness. More explanations are needed to demonstrate the contributions of different losses in the ablation study (Table 3).

---

> ### Author Rebuttal · Authors · 2023-08-09
>
> Thanks for your comments and suggestions. We will revise our paper accordingly.
>
> **C1: Lack demographics for the two testing datasets. And experiments can be performed on more populations for evaluation.**
>
> **A1**: We will provide detailed demographic information for all datasets used in our experiments.
>
> In the present study, we used multiple datasets obtained from ADNI-1, ADNI-2&GO, OASIS-1, and a Test-retest dataset. The ADNI-1 dataset consisted of 817 T1-weighted (T1w) brain MRIs from subjects aged 55 to 90, including normal controls (NC), mild cognitive impairment (MCI), and Alzheimer’s disease (AD), the ADNI-Go&2 dataset consisted of 200 T1w scans from subjects aged 60 to 87, including 100 NC and 100 AD subjests, the OASIS-1 dataset consisted of 413 T1w scans from subjects aged 18 to 96 years, including NC and AD subjects, and the Test-retest dataset consisted of 120 T1w scans from three subjects aged 26 to 31.
>
> Furthermore, we have also evaluated our method on HCP dataset with T1w scans from young adult aged 22 to 35. Due to time limit, we evaluated our method on a subset of randomly sampled 300 scans with a splitting of (training: 230, validation: 20, test: 50). The ASSD (mm) measures on L-Pial and L-WM surface reconstruction were 0.156 and 0.073, respectively.
>
> In summary, our method has been evaluated on MRI scans of subjects aged 18 to 96.
>
> We will further evaluate our method on scans of younger subjects in our ongoing studies. Particularly, we will evaluate our method on the dHCP dataset [4] that consists of 886 T2w scans from neonatal subjects (20 to 45 weeks).
>
> More details of the datasets used in our study can be found in their official websites. I will add the links to the supplementary materials.
>
> **C2: More explanations are needed to demonstrate the contributions of different losses in the ablation study (Table 3).**
>
> **A2**: Thanks for the instructive comments.
>
> We adopted multiple evaluation metrics to evaluate how different losses/components of our method contribute to the surface reconstruction in our experiments. In addition to commonly used metrics, such as Chamfer distance (CD), average symmetric surface distance (ASSD), and the 90th percentile Hausdorff distance (HD) that assess surface **_geometric accuracy_** but are not equipped to quantify the (spherical) topology of reconstructed surfaces, we also adopted the ratio of self-intersection faces (SIF) to measure the surface **_topological quality_** (Line 285). Overall, these complementary metrics enabled to quantify the contributions of different losses in the ablation study in terms of both accuracy and topological correctness.
>
> + The results of setting S4  (the model regularized with $L_{ch}$ alone) indicated that the model generated surfaces well matched to the ground truth data at the cost of high topological errors, particularly in highly curved regions, reflected by the result that the SIF was much worse on the pial surface than on the WM surface.
>
> + The results of setting S3 (the model regularized with $L_{ch}+L_{dist}$) indicated that the trajectory loss ($L_{dist}$) helped enforce the midthickness surface to reside between WM and pial surfaces, prevent surface intersection, and mitigate challenges in learning large deformations (Lines 144-149), reflected by the result that SIF was significantly reduced, particularly on the pial surface. A caveat is that the topology constraint might also yield marginally decreased accuracy in highly curved or deep sulci regions (Lines 326-328).
>
> + Our method jointly reconstructs both the inner and outer surfaces by deforming the midthickness surface inward and outward with two velocity fields (VFs). The cycle loss ($L_{cyc}$) was effective for promoting invertibility of the VFs, as shown in Fig 3. Such the invertibility also enabled to estimate the cortical thickness from the trajectory. Unlike $L_{cyc}$ working on each vertex, the symmetric similarity loss ($L_{ss}$) directly operated on the two VFs at the voxel level to further ensure the invertibility of these two VFs. The results of S2 ($L_{ch}+L_{dist}+L_{cyc}$) and S1 ($L_{ch}+L_{dist}+L_{cyc}+L_{ss}$) settings showed that surface quality was further improved. Geometric accuracy changes were possibly attributed to the smoothing effect of these loss terms.
>
> + Given that each vertex was deformed independently through sampled velocity from VFs, the normal consistency loss ($L_{nc}$) discouraged neighboring faces from forming sharp angles. These regularization terms improved both geometric accuracy and surface quality as indicated by the results of S0 setting using all losses above.

---

> > ### Comment · Reviewer_2ocD · 2023-08-18
> >
> > I have read other reviewers’ comments and the authors’ responses. Thank the authors for addressing my questions / concerns in the rebuttal. I would like to maintain my “Accept” score.

---

### Decision · Program_Chairs · 2023-09-21

**Decision:**

Accept (poster)

**Comment:**

In this paper, the authors propose a new method for cortical surface reconstruction. Most reviewers find the proposed method interesting and novel, with convincing experimental results. After a careful evaluation of the reviews and rebuttal, the AC agrees with these assessment and recommends acceptance.